# Learning with Admissibility: Robust Fuzzy Hashing for Cross-Modal Retrieval with Noisy Labels

Xincheng Sun [1 2 *]   Ruitao Pu [3 *]   Guangsi Shi [4]   Zhenwen Ren [5]   Peng Hu [3]   Yuan Sun [1]

## Abstract

Recently, cross-modal hashing (CMH) has garnered significant attention due to its low storage costs and high retrieval efficiency. Most existing CMH methods implicitly assume the availability of high-quality annotations, which is often violated in real-world scenarios as label noise inevitably arises from human errors or non-expert annotations. To cope with noisy supervision, current noise-robust CMH methods mainly follow two paradigms, i.e., noise separation and label smoothing. They often discard the predicted noisy instances or smooth discriminative signals to mitigate the impact of noisy labels. However, aggressive separation leads to reduced data utilization, while smoothing weakens the discriminative capability regarding the true distribution of clean instances. To address these limitations, we propose a novel Robust Fuzzy Cross-modal Hashing framework (RFCMH) that introduces fuzzy set theory to endow the labels with admissibility, thereby obtaining reliable discriminative supervision from noisy labels. Specifically, we first leverage possibility and necessity measures to model the noisy labels. Subsequently, we propose Fuzzy Admissibility Refinement (FAR) to dynamically calibrate supervision signals, thereby preventing the model from being misled by false positives. Furthermore, we introduce Dual-Granularity Structural Alignment (DGSA) to enforce both cross-modal alignment and instance-level uniformity,

ensuring stable and diverse representations. Extensive experiments on multiple benchmarks demonstrate that RFCMH achieves state-of-the-art retrieval performance. Code is available at https://github.com/XinchengSun/RFCMH.

## 1. Introduction

With the exponential growth of multimedia data, Cross-Modal Retrieval (CMR) (Qian et al., 2022; Zhou et al., 2023) has attracted significant attention for its ability to bridge the semantic gap between heterogeneous modalities. However, traditional CMR methods (Ge et al., 2023; Li et al., 2024; 2025a; Feng et al., 2025; 2026b;a) typically rely on high-dimensional real-valued representations, which incur prohibitive storage costs and search latencies in large-scale scenarios (Wen et al., 2023; Liu et al., 2025). To this end, Cross-Modal Hashing (CMH) has emerged as a promising solution (Su et al., 2025; Pu et al., 2025b; Su et al., 2026a; Li et al., 2026). By mapping high-dimensional data into compact binary codes, CMH could significantly reduce storage overhead and accelerate retrieval speed through bitwise XOR operations in the common Hamming space (Lan et al., 2025; Li et al., 2026).

Although existing CMH methods have obtained pleasing results, most of them rely on the implicit assumption that training data are perfectly annotated. In practice, label noise is ubiquitous due to non-expert or automated annotation processes (Kuznetsova et al., 2020; Song et al., 2022; Yin et al., 2025). They unconsciously overlook the impact of noisy labels. Such wrong supervision signals could mislead the model in CMH (Wang et al., 2024; 2026), thereby corrupting the semantics between heterogeneous modalities and resulting in the overfitting problem.

To resist the adverse impact of noisy labels, some robust CMH methods have been proposed, which can be roughly divided into two paradigms, i.e., noise separation and label smoothing. The former primarily focuses on distinguishing between clean and noisy instances, subsequently discarding or calibrating mislabeled instances to ensure reliable supervision. However, they heavily rely on the early prediction confidence of the model. As shown in Fig. 1 (a), in high

*Equal contribution [1]National Key Laboratory for Fundamental Algorithms and Models for Engineering Simulation, Sichuan University, Chengdu, China [2]Department of Computer Science and Technology, Heilongjiang University, Harbin, China [3]School of Computer Science, Sichuan University, Chengdu, China [4]Corporate Research Center, Midea Group, Shanghai, China [5]Southwest University of Science and Technology, Mianyang, China. Correspondence to: Yuan Sun <sun-yuan_work@163.com>.

*Proceedings of the 43rd International Conference on Machine Learning*, Seoul, South Korea. PMLR 306, 2026. Copyright 2026 by the author(s).

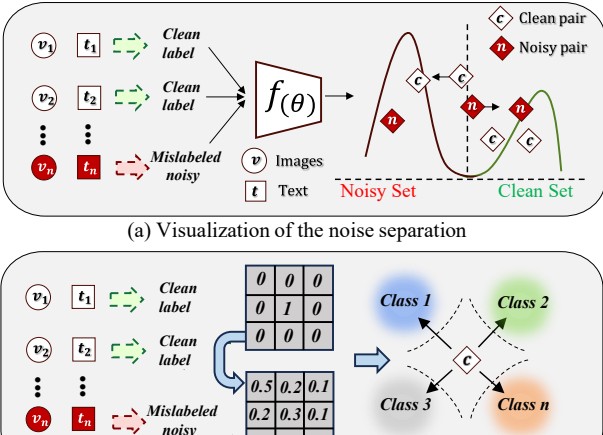

(a) Visualization of the noise separation

(b) Visualization of the label smoothing

*Figure 1.* (a) Noise separation paradigm, where a classifier $f(\theta)$ assigns a heuristic reliability score to each training instance (e.g., based on prediction confidence) and then hard-partitions the data into clean and noisy subsets. When label noise is high and instance semantics are ambiguous, the reliability scores of clean and noisy pairs become less separable and heavily overlap, leading to inevitable misclassification. (b) Label smoothing paradigm, where hard annotations are relaxed into soft distributions to mitigate noise interference. Although this soft supervision avoids aggressive sample removal, it weakens discriminative signals by associating instances with multiple categories, resulting in blurred decision boundaries and weakened class separability.

noise rate scenarios, due to the presence of instances with complex semantics, this makes the reliability determination difficult, which causes the losses or confidence distributions of clean and noisy instances to overlap indistinguishably. Consequently, the model often falls into a vicious cycle of clean instance scarcity and error accumulation during optimization. Moreover, overly aggressive separation strategies could lead to a sharp reduction in training data, resulting in a significant loss of available supervisory information. To alleviate the brittleness of hard noise separation, the label smoothing paradigm attempts to mitigate noise interference by relaxing hard targets into soft distributions, as depicted in Fig. 1 (b). In low-noise or nearly clean scenarios, this introduced uncertainty inhibits high-confidence predictions for correct labels, resulting in blurred decision boundaries. This inevitably weakens the model's discriminative capability regarding the true distribution of clean instances.

To address the aforementioned problems, we propose a novel Robust Fuzzy Cross-Modal Hashing (RFCMH) framework, which introduces fuzzy set theory to learn with admissibility from noisy labels, thereby obtaining reliable discriminative supervision. As shown in Fig. 2, our RFCMH consists of two stages, i.e., warm-up and fine-tuning. Specifically, during the warm-up stage, we model the membership of samples belonging to specific categories based on possibility and necessity measures. This process yields fuzzy admissibility measures aligned with category semantics.

Meanwhile, to reduce heterogeneous distribution discrepancies across modalities, we adopt an intra-instance contrast module to maximize the mutual information between representations of the same instance across modalities. In the fine-tuning stage, to prevent incorrect labels from degrading the accuracy of the admissibility measures, we propose a Fuzzy Admissibility Refinement (FAR) module. FAR alleviates the influence of unreliable labels by applying label smoothing to the admissibility distribution, effectively suppressing semantic shifts caused by noisy categories. After obtaining reliable category membership estimates during the warm-up stage, we further aim to enhance inter-class separability in the hash space. To this end, we propose a Dual-Granularity Structural Alignment (DGSA) strategy. DGSA enforces explicit inter-class structural constraints at the instance level, which enlarges the discriminative margins between samples from different categories while preserving multimodal semantic consistency. Overall, the main contributions of this paper are summarized as follows:

- This paper proposes a novel RFCMH framework to introduce fuzzy set theory into the field of cross-modal hashing. By assigning admissibility to noisy labels, our RFCMH could obtain reliable discriminative supervision signals to effectively overcome the performance degradation problem under noise interference.

- By integrating the measures of possibility and necessity, we propose FAR to explicitly suppress interference from error categories, thereby refining a more reliable soft supervision signal.

- Extensive experiments conducted on three widely used benchmark datasets validate the efficacy of our approach. Compared with 12 state-of-the-art baselines, RFCMH exhibits consistently superior robustness in the presence of label noise.

## 2. Related Work

Cross-modal hashing (CMH) aims to map heterogeneous modalities into a shared Hamming space, enabling efficient large-scale cross-modal retrieval with compact binary codes (Hu et al., 2022; Chen et al., 2026; Zhang et al., 2026). Existing CMH methods can be broadly categorized into unsupervised methods (Zhang et al., 2018; Su et al., 2019; Yu et al., 2021; Hu et al., 2022; Zhu et al., 2022; Cui et al., 2024) and supervised methods. Unsupervised methods primarily exploit cross-modal structural consistency or self-supervised cues to learn hash representations, circumventing the need for manual annotations. Nonetheless, the absence of explicit semantic supervision inevitably restricts the discriminability of the learned Hamming space, often resulting in performance bottlenecks for precise retrieval tasks. To

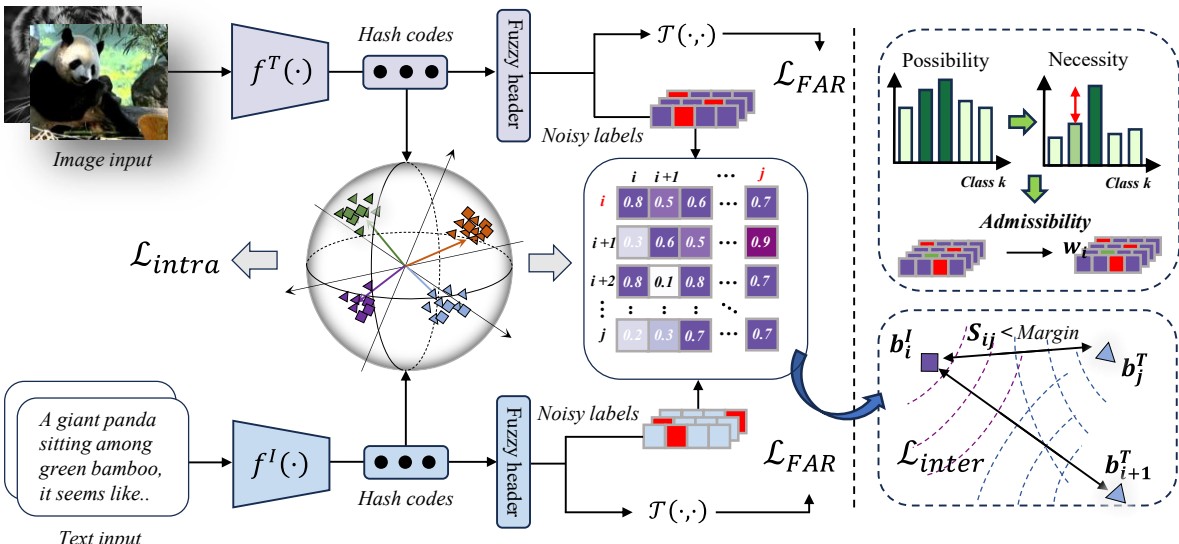

*Figure 2.* Illustration of RFCMH. Modality-specific hashing networks $f^I(\cdot)$ and $f^T(\cdot)$ encode paired image/text inputs into shared representations. A fuzzy semantic head produces category-wise membership scores, which are converted into admissibility via the possibility–necessity duality $\mathcal{T}(\cdot, \cdot)$ and optimized by $\mathcal{L}_{\text{FAR}}$ to provide reliable supervision under noisy labels. Training follows a progressive strategy: after the warm-up establishes coarse semantics, FAR further refines admissibility to suppress noisy categories, while DGSA constrains inter-instance relationships through intra-instance alignment $\mathcal{L}_{\text{intra}}$ and inter-instance separation $\mathcal{L}_{\text{inter}}$. The resulting representations yield noise-robust hash codes for cross-modal retrieval.

surpass these inherent limitations and bridge the semantic gap, the research focus has increasingly shifted toward supervised CMH (Tan et al., 2022; Sun et al., 2023; Zhang et al., 2022b;a; Huo et al., 2024; Liu et al., 2024; Sun et al., 2024a;b). By leveraging explicit labels, supervised methods can construct more discriminative hash codes that better reflect semantic similarities. However, the robustness of these methods in real-world scenarios is frequently undermined by imperfect supervision. In particular, label noise can severely corrupt the construction of similarity structures and exacerbate the quantization-related mismatch between continuous features and discrete codes.

To address this challenge, a series of noise-robust CMH methods have been developed (Wang et al., 2021; Yang et al., 2022; Wang et al., 2024; Pu et al., 2025c; Shu et al., 2024; Su et al., 2026b), often informed by broader research into noisy cross-modal learning beyond the hashing paradigm (Yang et al., 2025; Gan et al., 2024). Many of these approaches, however, hinge on hard sample separation (e.g., partitioning samples into clean versus noisy) or heuristic supervision reconstruction. In the early stages of training when representations have not yet converged, such hard decisions are easily perturbed and may become confounded, potentially propagating and accumulating errors during iterative optimization. Motivated by this, label smoothing has

been introduced to relax hard one-hot labels into real-valued targets, but inevitably weakens the original discriminative information (Xu et al., 2022; Huang et al., 2024).

However, both paradigms fail to explicitly assess whether noisy labels provide reliable support or exclusion for a given category during representation learning. How to extract reliable discriminative supervision from noisy labels remains underexplored, motivating us to revisit noise-robust cross-modal hashing through the lens of fuzzy set theory.

## 3. Method

### 3.1. Problem Formulation

To enhance clarity, we first introduce the fundamental definitions and notations for cross-modal hashing with noisy labels. Let $\mathcal{D} = \left\{ \left( \{\mathbf{x}_i^m\}_{m=1}^M, \mathbf{y}_i \right) \right\}_{i=1}^N$ denote a multi-modal dataset containing $N$ instances from $M$ modalities. Specifically, $\mathbf{x}_i^m$ represents the feature vector of the $m$-th modality for the $i$-th instance. In a bi-modal scenario, $m = 1$ and $m = 2$ correspond to image and text modalities, respectively. The associated label matrix is denoted as $\mathbf{Y} = [\mathbf{y}_1, \ldots, \mathbf{y}_N]^\top \in \mathbb{R}^{N \times K}$, where $K$ is the number of categories. Each $\mathbf{y}_i \in \{0, 1\}^K$ is a potentially noisy label vector: $y_{i,k} = 1$ if instance $i$ belongs to category $k$, and 0 otherwise. For notational convenience, let $t_i = \arg \max_{j \in \{1, \ldots, K\}} y_{ij}$

denote the index of the annotated category for instance $i$. Cross-modal hashing aims to learn modality-specific hash functions $\mathcal{H}^m(\cdot; \Theta^m)$, parameterized by $\Theta^m$, which map heterogeneous data into a shared $L$-bit Hamming space. During training, since directly optimizing discrete binary codes is NP-hard (Shen et al., 2015; Huo et al., 2024), we adopt a continuous relaxation and compute a binary-like representation via $\mathbf{h}_i^m = \tanh(\mathcal{H}^m(\mathbf{x}_i^m; \Theta^m))$. We further $\ell_2$-normalize $\mathbf{h}_i^m$ to obtain the shared representation $\mathbf{z}_i^m$, which is employed in subsequent objective functions. At inference, the final binary code is obtained as $\mathbf{b}_i^m = \text{sign}(\mathcal{H}^m(\mathbf{x}_i^m; \Theta^m)) \in \{-1, 1\}^L$, where $L$ denotes the code length.

### 3.2. Fuzzy Admissibility Refinement

Existing methods (Pu et al., 2025a) predominantly rely on deterministic point estimation, which inevitably leads to overconfident predictions on a certain class. In the presence of noisy labels, such overconfidence drives the model to overfit to erroneous supervision, thereby severely jeopardizing representation learning. To mitigate this, inspired by fuzzy set theory (Zadeh, 1965), we propose the Fuzzy Admissibility Refinement (FAR). FAR characterizes semantic conflicts arising from misannotations and multi-semantic ambiguity by leveraging the inherent capability of fuzzy logic, thereby effectively dampening the detrimental impact of label noise. Specifically, for a given sample $\mathbf{x}_i^m$, we first employ a possibility collection layer $\mathcal{P}(\cdot) \in \mathbb{R}^{L \times K}$ to estimate its possibility degrees across categories. This process can be formulated as $p_i^m = \{p_{i1}^m, p_{i2}^m, \ldots, p_{iK}^m\} = \sigma(\mathcal{P}(z_i^m))$, where $\sigma(\cdot)$ is the sigmoid activation function. Then, since a high possibility score does not necessarily imply semantic exclusivity (Duan et al., 2025b;a), the possibility measure alone is insufficient to assess supervision reliability under noisy annotations. For this issue, we introduce a label-aware necessity measure that explicitly accounts for category exclusiveness and enhances the reliability of supervision. For a sample $x_i^m$, its necessity measure with respect to category $k$ is defined as follows:

$$n_{ik}^m = \begin{cases} 1 - \max_{j \neq k} p_{ij}^m, & y_{ik} = 1, \\ p_{it_i}^m, & y_{ik} = 0, \end{cases} \quad (1)$$

where $p_{ij}^m$ denotes the possibility degree of the sample $x_i^m$ to category $j$. Under this measure, the necessity for a label-indicated category quantifies its semantic exclusivity against all alternatives, whereas the necessity for a non-label category reflects the dominance of the labeled category. For the target category, we incorporate $n_{ik}^m$ into the admissibility modeling in a weighted manner. To prevent over-suppression of the correct category under noisy annotations, we aggregate the necessity signals from the top-$\kappa$ non-target categories to form a robust necessity estimate for modeling semantic interference. Thus, the admissibility score $a_{ik}^m$ is defined as:

$$a_{ik}^m = \begin{cases} \frac{p_{ik}^m + \kappa n_{ik}^m}{1 + \kappa} & \text{if } y_{ik} = 1, \\ \frac{p_{ik}^m + n_{ik}^m}{2} & \text{if } y_{ik} = 0, \end{cases} \quad (2)$$

where $a_{ik}^m$ quantifies the extent to which the semantic signal for category $k$ remains acceptable under semantic exclusivity constraints. Since deep models prioritize clean/simple samples in the initial training phase (Liu et al., 2020), we directly align the admissibility scores with annotations to warm up the model, thereby obtaining a reliable admissibility distribution. The objective is defined as:

$$\mathcal{L}_{\text{FAR}} = \frac{1}{N} \sum_{m \in \{1,2\}} \sum_{i=1}^{N} \|\mathbf{a}_i^m - \mathbf{y}_i\|_2^2. \quad (3)$$

After completion of the warm-up stage, a reasonably reliable admissibility distribution could be obtained. Entering the fine-tuning stage, we build on this foundation to exploit the label-induced dominance structure encoded within the admissibility scores. This strategy regulates the supervision signal, thereby effectively alleviating the detrimental impact of noisy annotations. Specifically, for each instance $i$, let $\mathbf{a}_i^m = \{a_{i1}^m, \ldots, a_{iK}^m\}$ denote its category-wise admissibility scores under modality $m$. To synthesize a unified semantics, we first integrate the admissibility information across modalities by computing the cross-modal average: $\bar{\mathbf{a}}_i = \frac{1}{2}(\mathbf{a}_i^1 + \mathbf{a}_i^2)$. Subsequently, to evaluate the consistency between the label-induced dominance structure and the learned cross-modal semantics, we formulate a reliability weight $w_i$ as follows:

$$w_i = \sigma((\bar{a}_{i,t_i} - \max_{k \neq t_i} \bar{a}_{i,k} - \delta)/\tau), \quad (4)$$

where $\delta$ controls the conservativeness of reliability estimation, and $\tau$ is a temperature parameter. To explicitly determine whether a pair of instances should be pulled together or pushed apart in the shared semantic space, we define the discrimination indicator as: $\zeta_{ij} = \begin{cases} +1, & t_i = t_j \\ -1, & t_i \neq t_j, \end{cases}$. Finally, incorporating the reliability weights derived above, we impose a reliability-weighted margin constraint to facilitate robust representation learning:

$$\mathcal{L}_{\text{FAR}}^{\dagger} = \sum_{i \neq j} \min(w_i, w_j) \max(0, \gamma - \zeta_{ij}(\mathbf{z}_i^1)^{\top} \mathbf{z}_j^2), \quad (5)$$

where $\gamma$ is the predefined margin.

### 3.3. Dual-Granularity Structural Alignment

The basic goal of cross-modal retrieval is to bridge the inherent heterogeneity gap and effectively distinguish inter-class

samples. To carry this out, we propose the Dual-Granularity Structural Alignment (DGSA) module to simultaneously enhance the consistency of multimodal representations and augment inter-class discriminability. The DGSA module comprises two key components: intra-instance contrast and inter-instance contrast. Specifically, the intra-instance contrast component focuses on reinforcing multimodal consistency. It treats paired representations of the same instance as positives and the remaining instances as negatives (Chen et al., 2020; He et al., 2020), aiming to maximize the mutual information between positive pairs (Hu et al., 2021). To implement this, we first define the matching probabilities for the two retrieval directions as follows:

$$p_{ii}^{\text{I2T}} = \frac{\exp(S_{ii}/\tau)}{\sum_{j\in\mathcal{B}} \exp(S_{ij}/\tau)}, \quad p_{ii}^{\text{T2I}} = \frac{\exp(S_{ii}/\tau)}{\sum_{j\in\mathcal{B}} \exp(S_{ji}/\tau)}, \quad (6)$$

where $S_{ij}$ denotes the cosine similarity between the $i$-th image and the $j$-th text, and $\tau$ is the temperature parameter. However, using the standard log-loss $-\log p_{ii}^v$ to optimize near-binary hashing representations can be unstable. When the predicted probability is small, the gradient of the log-loss grows rapidly, leading to excessively large updates for poorly aligned pairs. To alleviate this issue, we replace the log-loss with a $q$-contrastive surrogate:

$$\mathcal{L}_{\text{intra}}^v = \frac{1}{B}\sum_{i=1}^{B} \ell_q\!\left(p_{ii}^v\right), \quad \ell_q(p) = \frac{(1-q)(1-p^q)}{q} + q(1-p), \quad (7)$$

where $q \in (0, 1]$. For this surrogate, as $p \to 0^+$, its gradient grows strictly slower than $-1/p$. Consequently, optimization is primarily driven by moderately aligned pairs, and extremely low-probability pairs no longer dominate the training dynamics, leading to more stable representations.

The inter-instance contrast component is to enlarge the separation between distinct instances by emphasizing margin-violating negatives. Specifically, for each anchor instance $(x_i^1, x_i^2)$, we enforce a strict constraint where the similarity $S_{ii}$ is encouraged to exceed any mismatched similarity by a margin $\gamma$, i.e., $S_{ii} \geq S_{ij} + \gamma$ for all $j \neq i$. To suppress easy negatives that already satisfy this constraint, we define a penalized similarity cost for each retrieval direction $v \in \{\text{I2T}, \text{T2I}\}$ as:

$$C_{ij}^v = \begin{cases} S_{ij}, & \text{if } S_{ij} > S_{ii} - \gamma, \\ S_{ij} - \Delta, & \text{otherwise}, \end{cases} \quad (8)$$

where $\Delta$ is a relaxation term that downweights easy negatives. For the T2I direction, $S_{ij}$ is replaced by $S_{ji}$ accordingly. Based on the penalized similarities, we aggregate competing negatives via a Log-Sum-Exp (LSE) formulation as follows:

$$\mathcal{L}_{\text{inter}}^v = \frac{1}{N}\sum_{i=1}^{N}\left[-S_{ii} + \tau\log\!\left(\sum_{j\neq i}\exp\!\left(C_{ij}^v/\tau\right)\right) + \gamma\right]. \quad (9)$$

---

**Algorithm 1** Optimization Procedure of RFCMH

1: **Input:** Training data $\mathcal{D}$, batch size $B$, epochs $N_e$, warm-up epoch $t_p$, hyper-parameters $\alpha, \kappa, \delta, \tau, q, \gamma, \Delta$.
2: **Output:** Optimized parameters $\Theta^1, \Theta^2$.
3: **for** $t = 1$ **to** $N_e$ **do**
4:     Sample a mini-batch $\{(\mathbf{x}_i^1, \mathbf{x}_i^2, \mathbf{y}_i)\}_{i=1}^{B}$.
5:     Compute representations $\mathbf{z}_i^m = \mathcal{H}^m(\mathbf{x}_i^m)$, $m \in \{1, 2\}$.
6:     Compute similarities $S_{ij} = (\mathbf{z}_i^1)^\top \mathbf{z}_j^2$.
7:     Compute possibility $p_i^m$, necessity $n_{ik}^m$, and admissibility $a_{ik}^m$ (i.e., Eq. (1)).
8:     **if** $t \leq t_p$ **then**
9:         Compute $\mathcal{L}_{\text{FAR}}$ (i.e., Eq. (3)) and $\mathcal{L}_{\text{intra}}^v$ (i.e., Eq. (7)).
10:        $\mathcal{L}_{\text{total}} \leftarrow \mathcal{L}_{\text{FAR}} + \alpha \sum_v \mathcal{L}_{\text{intra}}^v$.
11:     **else**
12:        Compute reliability $w_i$ (i.e., Eq. (4)).
13:        Compute $\mathcal{L}_{\text{FAR}}^\dagger$ (i.e., Eq. (5)) and $\mathcal{L}_{\text{DGSA}}$ (i.e., Eq. (10)).
14:        $\mathcal{L}_{\text{total}} \leftarrow \mathcal{L}_{\text{FAR}}^\dagger + \alpha\mathcal{L}_{\text{DGSA}}$.
15:     **end if**
16:     Update $\Theta^1, \Theta^2$ by minimizing $\mathcal{L}_{\text{total}}$.
17: **end for**

---

By emphasizing margin-violating negatives, $\mathcal{L}_{\text{inter}}^v$ explicitly enlarges inter-instance separation and complements the diagonal alignment enforced by $\mathcal{L}_{\text{intra}}^v$. The overall DGSA objective is given by:

$$\mathcal{L}_{\text{DGSA}} = \sum_{v\in\{\text{I2T},\text{T2I}\}} \left(\mathcal{L}_{\text{intra}}^v + \mathcal{L}_{\text{inter}}^v\right). \quad (10)$$

Notably, the intra-instance contrast item ($\mathcal{L}_{\text{intra}}^v$) is applied throughout both the warm-up and fine-tuning stages to maintain fundamental cross-modal consistency. In contrast, the inter-instance contrast ($\mathcal{L}_{\text{inter}}^v$) is just applied in the fine-tuning stage as it needs reliable discrimination boundaries.

### 3.4. Optimization Strategy

By integrating the aforementioned losses, the overall objective of RFCMH is formulated as:

$$\mathcal{L}_{\text{total}} = \begin{cases} \mathcal{L}_{\text{FAR}} + \alpha\mathcal{L}_{\text{intra}}, & t \leq t_p, \\ \mathcal{L}_{\text{FAR}}^\dagger + \alpha\mathcal{L}_{\text{DGSA}}, & t > t_p, \end{cases} \quad (11)$$

where $t$ is the current training epoch, $t_p$ is the warm-up epoch, and $\alpha$ is a hyperparameter. Finally, the proposed RFCMH minimizes the overall objective in Equation 11 to optimize network parameters in a batch-by-batch manner using gradient descent. The overall training procedure is summarized in Algorithm 1, with implementation settings provided in Section 4.3.

## 4. Experiment

### 4.1. Datasets

To verify the effectiveness of RFCMH, we conduct extensive experiments on three widely used datasets, i.e., **XMedia** (Peng et al., 2015), **INRIA-Websearch** (Krapac et al., 2010), and **XMediaNet** (Peng et al., 2018). We follow the dataset partitioning protocol of RSHNL (Pu et al., 2025c) for a fair comparison. Statistics of the three datasets are summarized in Tab 1, and more details of these datasets are provided in Appendix B due to space limitations.

### 4.2. Experimental Setup

To evaluate the performance of RFCMH and the compared methods, we conduct two common cross-modal retrieval tasks, i.e., image-to-text (I2T) and text-to-image (T2I). Similar to (Liu et al., 2026b;a), we adopt Mean Average Precision (mAP) to evaluate retrieval performance, which is a widely used evaluation metric. To comprehensively assess robustness, we consider symmetric label noise with different rates (i.e., 0.2, 0.4, 0.6, and 0.8), and report results with hash code lengths of 16, 32, 64, and 128 bits.

*Table 1.* Statistics of the experimental datasets.

| Datasets | Modality | Instances (Train / Val / Test) | Feature |
|---|---|---|---|
| XMedia | Image | 4,000 / 500 / 500 | 4,096-d CNN |
| | Text | 4,000 / 500 / 500 | 3,000-d BoW |
| INRIA-Websearch | Image | 9,000 / 1,332 / 4,366 | 4,096-d CNN |
| | Text | 9,000 / 1,332 / 4,366 | 1,000-d BoW |
| XMediaNet | Image | 32,000 / 4,000 / 4,000 | 4,096-d CNN |
| | Text | 32,000 / 4,000 / 4,000 | 300-d Doc2Vec |

### 4.3. Implementation Details

All experiments are implemented in PyTorch and are conducted on a single NVIDIA GeForce RTX 4090 GPU. During training, we optimize the networks using Adam (Kingma, 2014) with $\beta_1 = 0.9$, $\beta_2 = 0.999$, and a weight decay of $1 \times 10^{-4}$. The random seed is fixed at 10 for all experiments. Unless otherwise specified, we train the model for a maximum of 150 epochs with a mini-batch size of 64 and set the learning rate to $1 \times 10^{-4}$. Specifically, the modality-specific encoders ($\mathrm{ImgNN}$ and $\mathrm{TextNN}$) both adopt a 3-layer Multi-Layer Perceptron (MLP) architecture with two hidden layers of 4096 units and an output dimension of $L$ bits. For the proposed DGSA module, the temperature parameter is set to $\tau = 0.7$ and the robustness parameter in the $q$-contrastive objective is set to $q = 0.01$. The margin hyper-parameter $\gamma$ is fixed at 0.2, and the relaxation term $\Delta$ for easy-negative downweighting is set to 1.0. For the reliability weight $w_i$, the conservative threshold $\delta$ is set to 0.05. We adopt a progressive optimization strategy with the warmup switch epoch $t_p = 3$. All baselines are implemented using their released source code and are evaluated under the same retrieval protocol.

### 4.4. Compared Methods

For a comprehensive comparison, we adopt the following baselines. (1) **General baselines** include HMAH (Tan et al., 2022), MIAN (Zhang et al., 2022b), WASH (Zhang et al., 2022a), HCCH (Sun et al., 2023), and DECH (Li et al., 2025b). (2) **Noise-robust baselines** include CMMQ (Yang et al., 2022), DHRL (Shu et al., 2024), DHaPH (Huo et al., 2024), NRCH (Wang et al., 2024), NACD (Su et al., 2026b), RSHNL (Pu et al., 2025c), and SCBCH (Peng et al., 2025).

### 4.5. Comparison with State-of-the-Art Methods

The mAP results are reported in Tables 2 to 4. Besides, we plot precision-recall curves with 16 and 128 bits on the INRIA-Websearch dataset in Fig 3. From these results, the following key observations can be drawn:

- In low-noise scenarios, many methods exhibit strong performance. In this regime, we believe that the observed performance differences are mainly attributed to the quality of structural alignment, since the supervision is largely reliable and can sufficiently guide discriminative learning.

- As noise rates escalate, the performance of most existing methods deteriorates precipitously. In contrast, our proposed RFCMH achieves stable performance. This is because the FAR module effectively suppresses interference arising from erroneous categories by integrating both possibility and necessity measures, thereby enabling the extraction of reliable supervisory semantics even in highly noisy environments.

- Sample selection strategies often falter under conditions of high semantic complexity, falling behind even general baselines on datasets with a large number of categories (e.g., XMediaNet with 200 classes, as evidenced by the performance gap between NRCH and DECH). This phenomenon may stem from the fact that hard separation becomes brittle when semantic boundaries are ambiguous; the distinction between clean and noisy supervision blurs, and filtering risks discarding informative yet uncertain samples.

- As shown in the precision-recall curves, our proposed RFCMH consistently achieves superior precision at the same recall levels compared to SOTA methods, further indicating the effectiveness of RFCMH.

- In general, benefiting from the synergistic integration of the Fuzzy Admissibility Refinement (FAR) and Dual-Granularity Structural Alignment (DGSA), RFCMH effectively extracts reliable supervisory signals from noisy annotations while ensuring stable heterogeneous data alignment. Consequently,

*Table 2.* The mAP scores with different bit lengths on the **Xmedia** dataset under different noise rates. The best results are highlighted in **boldface** and the second-best are underlined.

| Task | Method | Noise Ref. | 0.2 | | | | 0.4 | | | | 0.6 | | | | 0.8 | | | |
|---|---|---|---|---|---|---|---|---|---|---|---|---|---|---|---|---|---|---|
| | | | 16 | 32 | 64 | 128 | 16 | 32 | 64 | 128 | 16 | 32 | 64 | 128 | 16 | 32 | 64 | 128 |
| I2T | HMAH | TMM'22 | 78.7 | 84.4 | 86.7 | 88.2 | 55.0 | 65.1 | 71.4 | 74.6 | 22.3 | 29.8 | 33.5 | 37.6 | 8.7 | 9.6 | 10.7 | 10.7 |
| | WASH | TKDE'23 | 80.7 | 85.9 | 87.1 | 87.7 | 75.6 | 79.4 | 81.3 | 82.1 | 44.8 | 53.8 | 57.4 | 59.6 | 13.9 | 16.3 | 17.2 | 19.2 |
| | MIAN | TKDE'23 | 18.1 | 29.6 | 34.5 | 35.4 | 12.4 | 16.2 | 17.7 | 16.2 | 10.7 | 9.9 | 11.5 | 11.1 | 7.6 | 7.7 | 7.1 | 8.0 |
| | HCCH | TMM'24 | 71.1 | 81.9 | 82.1 | 84.4 | 71.6 | 76.8 | 78.2 | 80.6 | 60.3 | 61.5 | 61.5 | 72.5 | 26.3 | 41.2 | 41.1 | 48.5 |
| | DECH | AAAI'25 | 6.9 | 76.4 | 81.1 | 81.6 | 7.8 | 56.0 | 60.0 | 68.3 | 6.5 | 27.2 | 31.3 | 38.7 | 6.3 | 8.4 | 11.6 | 19.7 |
| | CMMQ | CVPR'22 | 86.6 | 87.9 | 87.4 | 86.6 | 74.8 | 77.4 | 74.5 | 72.0 | 51.5 | 43.8 | 38.9 | 38.7 | 18.3 | 18.5 | 13.7 | 12.6 |
| | DHRL | TBD'24 | 11.0 | 39.3 | 86.7 | 90.5 | 9.9 | 66.0 | 84.2 | 86.6 | 9.4 | 37.7 | 69.6 | 74.4 | 6.6 | 8.9 | 37.6 | 43.8 |
| | DHaPH | TKDE'24 | 79.3 | 84.9 | 86.9 | 88.6 | 69.6 | 78.2 | 82.6 | 84.6 | 52.3 | 66.0 | 72.8 | 79.8 | 42.7 | 48.3 | 60.6 | 70.2 |
| | NRCH | MM'24 | 83.8 | 85.1 | 83.1 | 84.2 | 80.7 | 82.2 | 83.1 | 86.0 | 76.9 | 81.5 | 81.8 | 84.2 | 76.5 | 79.3 | 80.3 | 82.2 |
| | NACD | NeurIPS'25 | 81.8 | 85.3 | 87.6 | 87.2 | 78.4 | 85.3 | 86.9 | 86.3 | 78.6 | 84.5 | 85.1 | 84.6 | 75.1 | 82.7 | 83.4 | 84.3 |
| | RSHNL | AAAI'25 | **89.5** | _90.0_ | _90.9_ | _90.8_ | _89.7_ | _90.2_ | _91.1_ | _89.8_ | _83.0_ | _87.7_ | _88.9_ | _88.4_ | _81.7_ | _88.1_ | _87.6_ | _84.7_ |
| | SCBCH | AAAI'26 | 34.6 | 74.4 | 80.5 | 85.5 | 33.4 | 66.9 | 81.2 | 84.8 | 38.6 | 75.1 | 81.8 | 83.1 | 29.7 | 67.7 | 77.9 | 83.4 |
| | RFCMH | Ours | **89.5** | **91.4** | **92.2** | **91.3** | **89.8** | **90.7** | **91.4** | **90.7** | **89.3** | **90.1** | **90.5** | **91.2** | **88.1** | **89.7** | **89.9** | **90.2** |
| T2I | HMAH | TMM'22 | 77.4 | 84.0 | 85.9 | 88.0 | 53.2 | 65.0 | 71.0 | 73.9 | 23.1 | 30.5 | 34.1 | 38.9 | 8.9 | 10.0 | 11.5 | 11.4 |
| | WASH | TKDE'23 | 81.6 | 86.0 | 86.9 | 88.3 | 75.0 | 78.7 | 81.3 | 82.1 | 44.6 | 53.6 | 56.7 | 59.4 | 14.3 | 16.7 | 17.3 | 19.4 |
| | MIAN | TKDE'23 | 15.5 | 22.1 | 28.5 | 29.0 | 11.0 | 14.3 | 16.2 | 15.0 | 8.6 | 9.6 | 10.9 | 10.7 | 7.0 | 7.3 | 7.0 | 7.7 |
| | HCCH | TMM'24 | 69.5 | 80.7 | 80.0 | 84.0 | 70.0 | 75.7 | 76.7 | 80.0 | 58.1 | 60.2 | 58.5 | 72.6 | 26.2 | 40.8 | 41.2 | 48.2 |
| | DECH | AAAI'25 | 8.1 | 73.6 | 79.9 | 80.7 | 8.1 | 55.7 | 57.8 | 65.7 | 9.3 | 28.5 | 34.4 | 37.0 | 6.1 | 8.4 | 11.5 | 23.6 |
| | CMMQ | CVPR'22 | 85.1 | 83.4 | 82.0 | 78.2 | 70.8 | 74.4 | 67.6 | 64.4 | 45.2 | 36.0 | 33.2 | 43.5 | 14.4 | 14.0 | 13.2 | 10.2 |
| | DHRL | TBD'24 | 10.6 | 38.1 | 86.5 | _91.0_ | 10.1 | 65.5 | 83.5 | 86.5 | 10.7 | 39.3 | 68.6 | 72.1 | 8.2 | 8.3 | 39.9 | 45.5 |
| | DHaPH | TKDE'24 | 78.9 | 84.9 | 88.2 | 89.8 | 69.5 | 77.9 | 83.9 | 86.0 | 51.5 | 64.9 | 72.9 | 80.7 | 42.3 | 49.2 | 60.0 | 70.9 |
| | NRCH | MM'24 | 83.9 | 84.4 | 83.5 | 82.4 | 81.1 | 82.6 | 82.8 | 85.1 | 77.2 | 81.1 | 81.6 | 83.2 | 77.9 | 79.7 | 79.8 | 82.2 |
| | NACD | NeurIPS'25 | 81.4 | 85.5 | 88.5 | 88.6 | 80.1 | 84.0 | 87.4 | 87.0 | 77.9 | 84.9 | 86.0 | 86.1 | 75.6 | 82.0 | 84.2 | 85.2 |
| | RSHNL | AAAI'25 | _86.5_ | **89.4** | _91.0_ | 90.6 | _87.4_ | **90.2** | _90.3_ | _90.3_ | _82.6_ | _87.4_ | _88.9_ | _88.8_ | _79.7_ | _86.8_ | _86.7_ | _85.3_ |
| | SCBCH | AAAI'26 | 36.1 | 75.6 | 85.1 | 86.5 | 33.4 | 66.9 | 85.8 | 86.3 | 38.4 | 75.8 | 84.0 | 84.8 | 32.4 | 71.9 | 80.7 | 85.1 |
| | RFCMH | Ours | **87.6** | _89.3_ | **91.1** | **91.1** | **87.8** | _89.5_ | **90.5** | **91.4** | **87.7** | **88.8** | **89.8** | **91.0** | **86.1** | **89.1** | **88.9** | **89.6** |

our RFCMH achieves superior retrieval performance across varying noise rates. To further support the generality of RFCMH, we additionally provide supplementary results under the clean setting in Appendix C.

### 4.6. Ablation Study

To validate the contribution of individual components, we conduct an ablation study under the 0.6 noise rate, reporting the mean mAP for both I2T and T2I retrieval in Table 5. We compare the full model against four variants: RFCMH-1 (without warmup), RFCMH-2 (without $\mathcal{L}_{FAR}$), RFCMH-3 (without $\mathcal{L}_{DGSA}$), and RFCMH-4 (without the necessity component in FAR). All variants share identical training protocols for fair comparison. The results demonstrate that the full RFCMH achieves superior performance, confirming the synergistic efficacy of the proposed components: **(1)** The performance drop in RFCMH-1 highlights the critical role of warmup in stabilizing optimization during early training stages, particularly when representations are unconverged under heavy noise. **(2)** The degradation observed in RFCMH-2 validates the necessity of $\mathcal{L}_{FAR}$ for generating reliable soft supervision signals to guide learning. **(3)** The lower mAP of RFCMH-3 confirms that $\mathcal{L}_{DGSA}$ is essential for preserving cross-modal structural alignment and enhancing inter-instance discriminability in the Hamming space. **(4)** The decline in RFCMH-4 indicates that the necessity component is vital for refining supervision quality, further stabilizing performance under severe noise. Overall, the ablation results confirm that the proposed components are complementary and jointly enable stable retrieval.

### 4.7. Robustness Study

To study the robustness of our RFCMH, we conduct comparisons with three baselines (i.e., RSHNL, NRCH, and DECH) in terms of mAP versus training epochs. As shown in Fig 4, under the clean setting, different methods exhibit comparable trends. When the noise rate increases to $\eta = 0.6$, the gap becomes evident: DECH shows an initial improvement followed by a subsequent degradation, suggesting late-stage overfitting to noisy supervision, whereas RFCMH and RSHNL remain substantially more stable. Meanwhile, RFCMH attains a higher validation mAP earlier in training and continues to improve steadily thereafter, implying that the warmup strategy together with fuzzy learning helps establish reliable soft supervision in the early stage and facilitates robust convergence under high-noise settings.

### 4.8. Parameter Analysis

To investigate the sensitivity of two hyperparameters (i.e., $\alpha$ and $\kappa$), we conduct experiments with 128-bit codes on the INRIA-Websearch dataset. Specifically, we vary one parameter with different values while keeping the other fixed to evaluate its individual impact on retrieval performance. **(1)** As shown in Fig. 5(a), RFCMH is sensitive to $\alpha$: the performance peaks at a small value (around $\alpha = 0.1$) and drops noticeably when $\alpha$ increases beyond this range, suggesting that overly large $\alpha$ over-emphasizes the structural alignment term. **(2)** Fig. 5(b) shows that the mAP increases

*Table 3.* The mAP scores with different bit lengths on the **INRIA-Websearch** dataset under different noise rates. The best results are highlighted in **boldface** and the second-best are underlined.

| Task | Method | Noise Ref. | 0.2 | | | | 0.4 | | | | 0.6 | | | | 0.8 | | | |
|---|---|---|---|---|---|---|---|---|---|---|---|---|---|---|---|---|---|---|
| | | | 16 | 32 | 64 | 128 | 16 | 32 | 64 | 128 | 16 | 32 | 64 | 128 | 16 | 32 | 64 | 128 |
| I2T | HMAH | TMM'22 | 26.4 | 34.5 | 39.8 | 42.1 | 19.0 | 28.3 | 34.4 | 38.4 | 10.3 | 17.5 | 24.1 | 29.1 | 5.2 | 8.3 | 13.4 | 18.2 |
| | WASH | TKDE'23 | 31.5 | 38.0 | 43.5 | 46.2 | 26.2 | 33.0 | 38.5 | 42.1 | 16.9 | 22.1 | 27.9 | 31.5 | 4.7 | 7.5 | 10.9 | 13.7 |
| | MIAN | TKDE'23 | 2.7 | 2.7 | 2.2 | 2.8 | 2.7 | 1.7 | 2.8 | 2.8 | 2.8 | 2.4 | 1.7 | 1.7 | 2.8 | 2.3 | 1.8 | 1.4 |
| | HCCH | TMM'24 | 11.8 | 20.5 | 28.5 | 37.6 | 8.8 | 13.1 | 21.9 | 33.9 | 5.4 | 8.3 | 12.8 | 22.2 | 3.3 | 3.2 | 5.5 | 10.2 |
| | DECH | AAAI'25 | 25.2 | 36.3 | 38.8 | 46.3 | 11.8 | 17.6 | 21.7 | 28.6 | 1.9 | 5.3 | 7.0 | 10.2 | 1.4 | 2.1 | 2.6 | 3.0 |
| | CMMQ | CVPR'22 | 31.1 | 35.5 | 38.3 | 39.6 | 27.8 | 32.1 | 34.4 | 35.8 | 17.0 | 21.2 | 27.9 | 30.4 | 4.1 | 3.9 | 11.6 | 9.8 |
| | DHRL | TBD'24 | 2.8 | 4.2 | 33.6 | 33.8 | 2.6 | 2.8 | 23.8 | 24.3 | 2.7 | 2.7 | 12.6 | 8.3 | 2.8 | 3.0 | 4.9 | 6.2 |
| | DHaPH | TKDE'24 | 24.0 | 32.8 | 39.5 | 44.3 | 22.3 | 29.3 | 37.0 | 42.0 | 19.9 | 28.1 | 34.7 | 39.8 | 19.5 | 26.1 | 33.6 | 38.5 |
| | NRCH | MM'24 | 31.4 | 38.8 | 40.1 | 41.2 | 31.3 | 37.1 | 39.5 | 41.4 | 31.3 | 36.8 | 39.1 | 41.0 | 30.7 | 36.6 | 38.7 | 40.5 |
| | NACD | NeurIPS'25 | 29.6 | 34.5 | 38.6 | 40.7 | 28.7 | 34.5 | 38.4 | 40.4 | 28.1 | 34.0 | 37.9 | 39.8 | 29.6 | 33.9 | 38.0 | 39.4 |
| | RSHNL | AAAI'25 | 39.3 | 48.0 | 51.9 | 52.4 | 37.9 | 45.8 | 50.3 | 51.6 | 31.2 | 38.3 | 47.6 | 49.0 | 28.3 | 38.2 | 41.8 | 42.9 |
| | SCBCH | AAAI'26 | 6.7 | 20.0 | 28.6 | 40.5 | 5.1 | 11.4 | 31.4 | 38.6 | 5.8 | 14.7 | 35.0 | 35.5 | 5.0 | 22.3 | 29.8 | 36.1 |
| | RFCMH | Ours | **46.2** | **51.9** | **54.5** | **55.2** | **45.9** | **51.2** | **53.4** | **54.7** | **43.2** | **49.7** | **53.3** | **54.9** | **41.2** | **47.7** | **50.6** | **52.0** |
| T2I | HMAH | TMM'22 | 25.0 | 34.5 | 40.8 | 43.9 | 18.2 | 28.3 | 34.9 | 39.7 | 9.9 | 17.1 | 24.4 | 29.4 | 4.7 | 7.9 | 13.1 | 17.7 |
| | WASH | TKDE'23 | 30.8 | 38.4 | 44.8 | 47.8 | 25.1 | 32.7 | 39.3 | 43.3 | 15.8 | 21.7 | 27.8 | 31.9 | 4.2 | 7.2 | 10.6 | 13.3 |
| | MIAN | TKDE'23 | 1.2 | 1.2 | 1.2 | 1.2 | 1.2 | 1.2 | 1.2 | 1.2 | 1.2 | 1.2 | 1.2 | 1.2 | 1.2 | 1.2 | 1.2 | 1.2 |
| | HCCH | TMM'24 | 12.7 | 23.6 | 35.0 | 42.5 | 9.4 | 17.0 | 29.4 | 39.0 | 5.9 | 10.8 | 19.7 | 29.3 | 3.0 | 3.6 | 7.1 | 13.1 |
| | DECH | AAAI'25 | 26.7 | 39.8 | 46.4 | 49.6 | 11.9 | 18.6 | 23.6 | 29.1 | 1.9 | 5.8 | 8.7 | 11.6 | 1.6 | 2.1 | 2.7 | 3.0 |
| | CMMQ | CVPR'22 | 31.4 | 36.6 | 38.5 | 39.0 | 27.3 | 32.3 | 34.5 | 34.1 | 16.2 | 20.6 | 27.3 | 30.4 | 4.3 | 3.4 | 10.7 | 8.5 |
| | DHRL | TBD'24 | 2.7 | 4.2 | 32.3 | 33.7 | 2.7 | 2.6 | 23.5 | 24.0 | 2.8 | 2.6 | 13.2 | 7.7 | 2.7 | 2.9 | 4.8 | 7.1 |
| | DHaPH | TKDE'24 | 22.4 | 32.8 | 40.7 | 45.7 | 21.0 | 29.3 | 37.6 | 43.1 | 18.6 | 28.4 | 35.4 | 40.8 | 18.6 | 25.6 | 33.8 | 39.5 |
| | NRCH | MM'24 | 30.4 | 39.8 | 41.4 | 43.2 | 30.8 | 37.8 | 40.6 | 42.7 | 31.0 | 37.2 | 40.4 | 42.0 | 29.8 | 37.2 | 39.6 | 41.9 |
| | NACD | NeurIPS'25 | 28.3 | 35.3 | 39.6 | 41.6 | 27.6 | 35.0 | 39.2 | 41.1 | 26.8 | 34.4 | 38.6 | 40.7 | 27.8 | 34.1 | 38.7 | 40.4 |
| | RSHNL | AAAI'25 | 38.2 | 47.9 | 52.1 | 53.3 | 36.2 | 45.9 | 50.3 | 52.3 | 30.2 | 37.7 | 47.9 | 49.8 | 30.7 | 38.0 | 42.0 | 42.9 |
| | SCBCH | AAAI'26 | 6.5 | 21.5 | 29.9 | 42.9 | 5.4 | 11.5 | 31.6 | 36.8 | 5.4 | 15.5 | 35.1 | 36.8 | 5.8 | 23.2 | 32.0 | 37.5 |
| | RFCMH | Ours | **45.1** | **50.8** | **54.5** | **55.6** | **45.1** | **50.9** | **53.7** | **55.2** | **42.6** | **49.5** | **52.7** | **56.0** | **39.1** | **47.0** | **50.8** | **51.9** |

*Table 4.* The mAP scores with different bit lengths on the **XMediaNet** dataset under different noise rates. The best results are highlighted in **boldface** and the second-best are underlined.

| Task | Method | Noise Ref. | 0.2 | | | | 0.4 | | | | 0.6 | | | | 0.8 | | | |
|---|---|---|---|---|---|---|---|---|---|---|---|---|---|---|---|---|---|---|
| | | | 16 | 32 | 64 | 128 | 16 | 32 | 64 | 128 | 16 | 32 | 64 | 128 | 16 | 32 | 64 | 128 |
| I2T | HMAH | TMM'22 | 2.9 | 3.6 | 5.9 | 10.9 | 1.3 | 1.5 | 1.9 | 3.9 | 1.0 | 1.0 | 1.3 | 2.2 | 1.0 | 1.1 | 1.3 | 1.7 |
| | WASH | TKDE'23 | 8.0 | 15.1 | 24.2 | 34.0 | 7.0 | 13.1 | 21.2 | 30.6 | 4.9 | 8.5 | 14.4 | 21.8 | 2.0 | 3.1 | 4.7 | 6.9 |
| | MIAN | TKDE'23 | 0.8 | 1.5 | 1.8 | 2.5 | 0.8 | 1.2 | 1.3 | 1.8 | 0.7 | 0.9 | 1.0 | 1.2 | 0.7 | 0.8 | 0.8 | 0.9 |
| | HCCH | TMM'24 | 1.6 | 2.0 | 4.8 | 15.0 | 1.4 | 1.5 | 3.4 | 12.1 | 1.3 | 1.4 | 2.2 | 5.8 | 0.8 | 0.9 | 1.1 | 1.9 |
| | DECH | AAAI'25 | 1.2 | 36.3 | 43.2 | 31.5 | 1.0 | 7.7 | 23.7 | 30.4 | 0.8 | 3.0 | 23.1 | 30.6 | 0.8 | 2.0 | 22.9 | 32.7 |
| | CMMQ | CVPR'22 | / | / | / | / | / | / | / | / | / | / | / | / | / | / | / | / |
| | DHRL | TBD'24 | 0.7 | 0.7 | 0.9 | 15.2 | 0.7 | 0.7 | 1.0 | 13.2 | 0.7 | 0.7 | 2.7 | 6.3 | 0.7 | 0.7 | 0.7 | 1.3 |
| | DHaPH | TKDE'24 | 9.8 | 15.9 | 23.4 | 28.5 | 9.1 | 15.8 | 22.6 | 27.6 | 9.2 | 15.1 | 21.9 | 27.5 | 8.8 | 15.2 | 21.5 | 27.3 |
| | NRCH | MM'24 | 9.5 | 14.2 | 17.5 | 20.3 | 9.1 | 13.7 | 17.2 | 20.3 | 8.9 | 13.2 | 16.7 | 19.7 | 8.5 | 13.2 | 16.4 | 19.5 |
| | NACD | NeurIPS'25 | 2.0 | 6.9 | 7.9 | 7.8 | 1.9 | 6.7 | 8.0 | 7.7 | 1.6 | 6.7 | 8.4 | 7.6 | 1.0 | 6.6 | 7.7 | 7.6 |
| | RSHNL | AAAI'25 | 35.3 | 43.5 | 47.5 | 48.6 | 35.5 | 42.1 | 47.0 | 45.8 | 33.2 | 41.8 | 46.2 | 45.0 | 28.2 | 39.6 | 44.9 | 38.9 |
| | SCBCH | AAAI'26 | 3.8 | 11.4 | 16.3 | 28.0 | 0.7 | 8.3 | 14.4 | 25.0 | 0.7 | 7.5 | 17.8 | 24.6 | 0.7 | 10.1 | 15.1 | 26.8 |
| | RFCMH | Ours | **40.9** | **54.4** | **52.7** | **55.7** | **41.2** | **50.5** | **52.3** | **54.5** | **39.5** | **49.5** | **53.3** | **54.2** | **30.2** | **48.9** | **50.4** | **54.0** |
| T2I | HMAH | TMM'22 | 3.5 | 4.4 | 6.6 | 12.1 | 1.6 | 1.8 | 2.3 | 4.4 | 1.1 | 1.2 | 1.5 | 2.5 | 1.0 | 1.2 | 1.4 | 1.8 |
| | WASH | TKDE'23 | 10.3 | 17.3 | 25.8 | 35.5 | 8.9 | 15.2 | 22.8 | 32.2 | 6.3 | 10.2 | 15.9 | 23.4 | 2.4 | 3.6 | 5.4 | 7.8 |
| | MIAN | TKDE'23 | 0.8 | 1.2 | 1.2 | 1.8 | 0.7 | 1.0 | 1.0 | 1.3 | 0.7 | 0.8 | 0.8 | 1.0 | 0.7 | 0.7 | 0.7 | 0.8 |
| | HCCH | TMM'24 | 1.8 | 1.5 | 2.1 | 9.0 | 1.4 | 1.2 | 1.7 | 12.1 | 1.5 | 1.1 | 1.3 | 1.8 | 0.8 | 0.9 | 1.0 | 1.2 |
| | DECH | AAAI'25 | 1.6 | 16.2 | 23.1 | 34.5 | 1.0 | 8.9 | 24.7 | 31.9 | 0.8 | 3.6 | 23.9 | 32.7 | 0.8 | 2.3 | 23.5 | 34.1 |
| | CMMQ | CVPR'22 | / | / | / | / | / | / | / | / | / | / | / | / | / | / | / | / |
| | DHRL | TBD'24 | 0.7 | 0.8 | 0.9 | 19.8 | 0.8 | 0.9 | 1.0 | 15.1 | 0.7 | 0.8 | 3.1 | 8.7 | 0.8 | 0.8 | 0.8 | 1.5 |
| | DHaPH | TKDE'24 | 11.0 | 18.0 | 26.3 | 32.6 | 10.5 | 17.3 | 25.3 | 31.6 | 10.3 | 16.8 | 24.8 | 31.2 | 9.6 | 17.1 | 24.0 | 30.7 |
| | NRCH | MM'24 | 10.6 | 16.2 | 20.1 | 23.6 | 10.4 | 15.5 | 19.5 | 23.7 | 10.1 | 14.9 | 19.0 | 22.8 | 9.6 | 14.8 | 18.5 | 22.7 |
| | NACD | NeurIPS'25 | 2.1 | 7.9 | 9.9 | 10.1 | 2.0 | 7.8 | 10.0 | 10.0 | 1.7 | 7.8 | 10.0 | 10.0 | 1.8 | 7.7 | 9.9 | 10.0 |
| | RSHNL | AAAI'25 | 35.3 | 42.5 | 46.5 | 47.8 | 35.0 | 41.6 | 46.9 | 46.6 | 33.3 | 41.3 | 46.4 | 46.1 | 29.3 | 39.9 | 45.1 | 40.6 |
| | SCBCH | AAAI'26 | 3.3 | 9.9 | 15.4 | 27.6 | 0.7 | 7.0 | 14.1 | 24.6 | 0.7 | 4.4 | 17.4 | 24.7 | 0.7 | 9.5 | 14.5 | 26.6 |
| | RFCMH | Ours | **40.6** | **53.9** | **53.2** | **55.2** | **40.8** | **50.5** | **54.5** | **55.2** | **38.9** | **49.5** | **53.7** | **54.9** | **32.0** | **49.2** | **51.2** | **54.1** |

rapidly when $\kappa$ moves from 0 to 1 and then remains nearly unchanged for larger $\kappa$, with the best performance achieved around $\kappa \in [3, 4]$. This trend supports the effectiveness of FAR in suppressing semantic interference and constructing robust necessity through the possibility–necessity duality. Compared with using admissibility derived solely from possibility scores (i.e., $\kappa = 0$), incorporating necessity-aware interference suppression (i.e., $\kappa = 1$) produces a clear gain by enforcing the label-induced dominance structure between the target category and its competitors. Moreover, under noisy supervision, increasing $\kappa$ strengthens the contribution of the aggregated top-$\kappa$ non-target necessity in admissibility, which smooths semantic interference and alleviates the over-suppression of the true category.

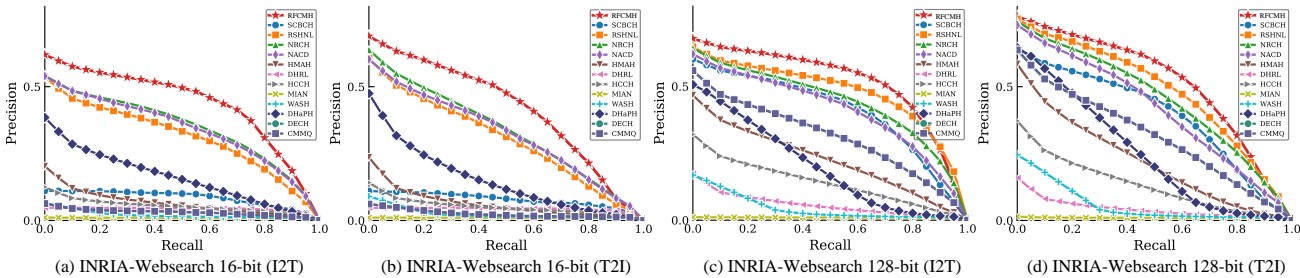

*Figure 3.* Precision–recall curves with different bit lengths on the INRIA-Websearch dataset under a noise rate of 0.8.

*Table 5.* Ablation study under the 0.6 noise rate.

| Datasets | XMedia | | | | | XMediaNet | | | | | INRIA-Websearch | | | | |
|---|---|---|---|---|---|---|---|---|---|---|---|---|---|---|---|
| Method \ Bit | 16 | 32 | 64 | 128 | Sum | 16 | 32 | 64 | 128 | Sum | 16 | 32 | 64 | 128 | Sum |
| RFCMH-1 | 87.6 | 88.5 | 89.8 | 89.8 | 355.7 | 36.6 | 44.8 | 48.4 | 50.3 | 180.1 | 42.7 | 47.0 | 49.2 | 48.2 | 187.1 |
| RFCMH-2 | 85.9 | 88.3 | 89.6 | 89.4 | 353.2 | 28.5 | 49.4 | 52.9 | 53.3 | 184.1 | 38.8 | 43.3 | 44.0 | 41.9 | 168.0 |
| RFCMH-3 | 36.9 | 41.4 | 52.4 | 64.7 | 195.4 | 1.1 | 5.6 | 10.8 | 20.4 | 37.9 | 9.9 | 15.4 | 24.1 | 29.6 | 79.0 |
| RFCMH-4 | 87.7 | 88.7 | 89.4 | 89.9 | 355.7 | 38.5 | 48.9 | 52.5 | **54.9** | 194.8 | **42.9** | 47.5 | 49.4 | 48.2 | 188.0 |
| RFCMH (Ours) | **88.5** | **89.5** | **90.2** | **91.1** | **359.3** | **39.2** | **49.5** | **53.5** | 54.6 | **196.8** | **42.9** | **49.6** | **53.0** | **55.5** | **201.0** |

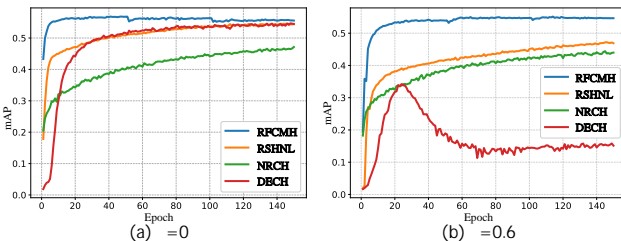

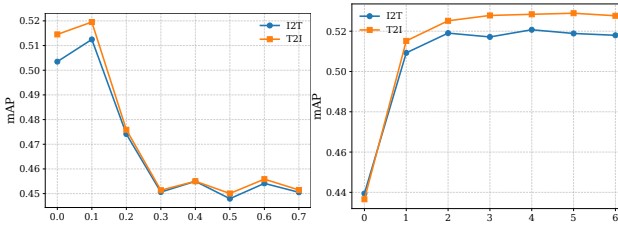

*Figure 4.* Mean validation mAP scores with 128-bit hash codes on the INRIA-Websearch dataset under different noise rates $\eta$.

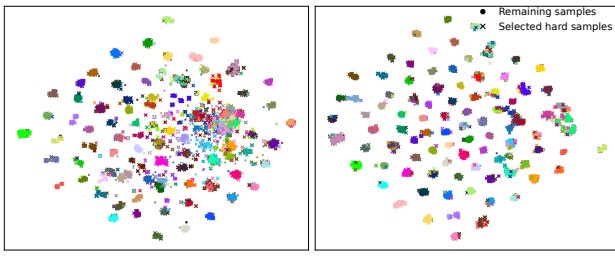

*Figure 5.* Parameter analysis on the INRIA-Websearch dataset under noise rate $\eta = 0.8$.

### 4.9. *t*-SNE Visualization

We employ t-SNE to visualize the feature embeddings of the INRIA-Websearch dataset under the 0.6 noise rate, as shown in Fig. 6. The results demonstrate: **(1)** Distinct semantic categories occupy separate regions, indicating that both NRCH and RFCMH successfully capture discriminative information. **(2)** Crucially, we visualize the separation of samples identified as noisy versus clean. In NRCH, samples assigned to the noisy cluster suffer from clustering collapse, aggregating into a dense, indistinguishable region. In contrast, RFCMH exhibits no such collapse; it maintains clear structural separation even for uncertain samples. This phenomenon further corroborates the robustness of RFCMH

*Figure 6.* t-SNE visualization of training representations on INRIA-Websearch.

in learning discriminative representations under noisy supervision.

## 5. Conclusion

In this paper, we propose a novel Robust Fuzzy Cross-modal Hashing (RFCMH) framework to address the challenge of noisy supervision. RFCMH consists of two key modules: Fuzzy Admissibility Refinement (FAR) and Dual-Granularity Structural Alignment (DGSA). Specifically, FAR couples category-wise possibility estimates with label-aware necessity to derive an admissibility distribution with explicit semantic exclusion, enabling reliable supervision refinement. DGSA enhances robustness by jointly enforcing intra-instance cross-modal consistency and discriminative separation with stabilized contrastive objectives on near-binary representations, thereby preserving a well-structured embedding geometry in the shared Hamming space. Experiments on multiple benchmarks and code lengths under different noise rates show RFCMH consistently outperforms strong baselines in noisy settings, with particularly pronounced gains at high noise levels.

## Acknowledgements

This work was supported by the Foundation Enhancement Program Project (Technology Field Fund) (Grant No. 2025-JCJQ-JJ-0686), the National Social Science Foundation of China (Grant No. 24BGL131), the Sichuan Science and Technology Planning Project (Grant Nos. 2026NS-FSC1480, 2024NSFSC0521), and the Luzhou City School-Local Enterprise-Academy Science and Technology Cooperation Project (Grant No. 2024XDY200).

## Impact Statement

This paper presents work whose goal is to advance the field of Machine Learning. There are many potential societal consequences of our work, none of which we feel must be specifically highlighted here.

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

# APPENDIX

This document provides supplementary materials, including theoretical foundations, experimental protocols, and extensive empirical evaluations, to support the findings of the main paper. The contents are organized as follows:

- **Appendix A: Theoretical Derivations and Proofs.** In this section, we provide the complete mathematical foundations for the propositions and loss functions introduced in Section 3. This includes, but is not limited to, gradient analysis and consistency proofs for our robust alignment objectives.

- **Appendix B: Experimental Details.** In this section, we provide additional experimental details for the feature extraction protocols, compared baselines, and implementation settings used in our experiments.

- **Appendix C: Extended Empirical Results and Analysis.** We present a broad range of supplementary experiments to further validate our method. This includes detailed ablation studies across various components, parameter sensitivity investigations, and qualitative visualizations of the cross-modal retrieval process.

- **Appendix D: Limitations and Future Work.** We provide a critical discussion of the current limitations of our approach and outline potential directions for future research.

## A. Theoretical Derivations and Proofs

### A.1. Gradient Robustness of the Intra-instance $q$-Contrastive Objective

**Proposition A.1.** *The gradient of the proposed intra-instance loss $\mathcal{L}_{\text{intra}}^{v}$ grows asymptotically slower than that of the standard cross-entropy loss used in InfoNCE, thereby suppressing the influence of outliers.*

*Proof.* Without loss of generality, let $p$ denote the probability of a positive cross-modal pair being correctly identified, as defined in Equation (7). The individual loss for a single instance is

$$\ell(p) = \frac{(1-q)(1-p^q)}{q} + q(1-p), \qquad q \in (0, 1]. \tag{12}$$

The first-order derivative with respect to $p$ is

$$\begin{aligned}
\frac{d\ell(p)}{dp} &= \frac{1-q}{q} \cdot (-qp^{q-1}) - q \\
&= -(1-q)p^{q-1} - q.
\end{aligned} \tag{13}$$

In contrast, the cross-entropy form of the InfoNCE objective is given by $\ell_{\text{ce}}(p) = -\log p$, whose gradient is

$$\frac{d\ell_{\text{ce}}(p)}{dp} = -\frac{1}{p}. \tag{14}$$

When an instance corresponds to a hard or noisy cross-modal pair, the predicted probability $p$ approaches 0. As $p \to 0$,

1. $\lim_{p \to 0} \left| \frac{d\ell_{\text{ce}}(p)}{dp} \right| = \infty$, indicating that extremely small probabilities can generate disproportionately large gradients.

2. For the proposed $q$-loss, since $q \in (0, 1]$, the gradient magnitude is governed by $p^{q-1} = 1/p^{1-q}$, which grows strictly slower than $1/p$ because $1 - q < 1$.

3. In the special case $q = 1$, the gradient reduces to the constant $-1$, which is bounded for all $p$.

Therefore, the formulation in Equation (7) reduces the optimization dominance of extremely small-$p$ outliers and suppresses the influence of unreliable cross-modal matches, leading to more stable representation learning. □

*Table 6.* The mAP scores with different bit lengths on three datasets. The best results are highlighted in **boldface** and the second-best are underlined. The column **Sum** indicates the total score of all hash lengths.

| Task | Method | Datasets Ref. | Xmedia | | | | | INRIA-Websearch | | | | | XmediaNet | | | | |
|---|---|---|---|---|---|---|---|---|---|---|---|---|---|---|---|---|---|
| | | | 16 | 32 | 64 | 128 | Sum | 16 | 32 | 64 | 128 | Sum | 16 | 32 | 64 | 128 | Sum |
| I2T | WASH | TKDE'23 | 88.2 | 87.4 | 89.8 | 90.0 | 355.4 | 33.4 | 40.1 | 45.6 | 48.0 | 167.1 | 8.6 | 16.3 | 24.9 | 35.5 | 85.3 |
| | HCCH | TMM'24 | 79.1 | 78.2 | 52.0 | 77.5 | 286.8 | 14.2 | 21.7 | 31.6 | 30.7 | 98.2 | 2.2 | 3.3 | 6.8 | 18.6 | 30.9 |
| | HMAH | TMM'22 | 86.0 | 89.2 | 90.0 | 90.3 | 355.5 | 33.6 | 43.1 | 47.6 | 46.2 | 170.5 | 7.8 | 13.0 | 18.5 | 25.7 | 65.0 |
| | CMMQ | CVPR'22 | 81.6 | 86.0 | 90.1 | 90.3 | 348.0 | 32.1 | 32.4 | 46.0 | 48.7 | 159.2 | 3.2 | 3.8 | 7.9 | 17.6 | 32.5 |
| | DHRL | TBD'24 | 65.4 | 75.2 | 83.2 | 84.5 | 308.3 | 2.7 | 3.5 | 37.8 | 43.1 | 87.1 | 1.1 | 20.1 | 26.0 | 34.1 | 81.3 |
| | DHaPH | TKDE'24 | 88.1 | 90.4 | 91.8 | **93.2** | 363.5 | 24.8 | 34.3 | 41.2 | 45.8 | 146.1 | 6.1 | 10.7 | 16.3 | 22.5 | 55.6 |
| | RSHNL | AAAI'25 | **90.5** | 91.2 | **92.3** | 91.9 | **365.9** | 36.2 | 45.2 | 50.0 | 51.9 | 183.3 | 32.4 | 41.4 | 47.6 | 48.6 | 170.0 |
| | DECH | AAAI'25 | 85.5 | 89.7 | 90.9 | 91.8 | 357.9 | 30.9 | 44.5 | 50.1 | 52.0 | 177.5 | 6.9 | 25.1 | 42.3 | 49.9 | 124.2 |
| | NRCH | MM'24 | 86.5 | 89.5 | 90.1 | 91.2 | 357.3 | 30.6 | 37.0 | 40.7 | 46.1 | 154.4 | 7.4 | 12.9 | 16.7 | 18.7 | 55.7 |
| | NACD | NeurIPS'25 | 85.2 | 89.4 | 90.9 | 90.7 | 356.2 | 29.0 | 35.1 | 39.1 | 41.0 | 144.2 | 2.1 | 7.5 | 8.9 | 12.3 | 30.8 |
| | SCBCH | AAAI'26 | 41.3 | 67.9 | 82.9 | 82.3 | 274.4 | 5.8 | 18.5 | 31.9 | 42.6 | 98.8 | 1.1 | 5.8 | 13.6 | 23.5 | 44.0 |
| | RFCMH | Ours | 90.2 | **91.3** | 92.1 | 91.4 | 365.0 | **48.4** | **52.0** | **54.0** | **55.8** | **210.2** | **43.7** | **55.5** | **55.9** | **56.0** | **211.1** |
| T2I | WASH | TKDE'23 | 87.4 | 88.4 | 89.0 | 89.2 | 354.0 | 32.5 | 41.2 | 47.6 | 50.5 | 171.8 | 10.7 | 18.6 | 26.8 | 36.8 | 92.9 |
| | HCCH | TMM'24 | 79.7 | 77.4 | 56.8 | 74.5 | 288.4 | 13.2 | 24.9 | 37.5 | 26.7 | 102.3 | 2.5 | 2.4 | 3.8 | 15.9 | 24.6 |
| | HMAH | TMM'22 | 85.3 | 88.9 | 89.6 | 90.5 | 354.3 | 33.3 | 44.3 | 49.6 | 48.2 | 175.4 | 9.1 | 14.5 | 20.0 | 27.2 | 70.8 |
| | CMMQ | CVPR'22 | 74.1 | 80.4 | 90.6 | 90.6 | 335.7 | 30.9 | 30.4 | 48.6 | 51.6 | 161.5 | 4.7 | 6.2 | 10.4 | 20.2 | 41.5 |
| | DHRL | TBD'24 | 65.0 | 78.7 | 83.6 | 85.2 | 312.5 | 2.9 | 2.6 | 38.3 | 44.2 | 88.0 | 1.0 | 18.7 | 25.9 | 33.6 | 79.2 |
| | DHaPH | TKDE'24 | **89.5** | **91.6** | **91.8** | **92.4** | **365.3** | 23.6 | 34.5 | 42.7 | 47.2 | 148.0 | 6.6 | 11.6 | 17.6 | 24.0 | 59.8 |
| | RSHNL | AAAI'25 | 88.9 | 90.5 | 91.6 | 91.9 | 362.9 | 35.6 | 45.7 | 51.1 | 53.5 | 185.9 | 33.3 | 41.6 | 47.7 | 49.3 | 171.9 |
| | DECH | AAAI'25 | 82.0 | 88.8 | 90.5 | 90.8 | 352.1 | 30.0 | 44.8 | 52.0 | 53.3 | 180.1 | 9.6 | 26.3 | 42.8 | 51.1 | 129.8 |
| | NRCH | MM'24 | 87.7 | 89.9 | 90.0 | 90.2 | 357.8 | 31.9 | 35.7 | 37.0 | 40.0 | 144.6 | 6.3 | 10.6 | 15.1 | 16.9 | 48.9 |
| | NACD | NeurIPS'25 | 84.1 | 89.1 | 91.1 | 91.3 | 355.6 | 27.3 | 35.8 | 40.0 | 42.0 | 145.1 | 2.1 | 8.3 | 10.2 | 13.4 | 34.0 |
| | SCBCH | AAAI'26 | 33.3 | 71.4 | 86.5 | 86.4 | 277.6 | 7.0 | 18.9 | 34.0 | 45.8 | 105.7 | 1.2 | 6.2 | 12.5 | 23.3 | 43.2 |
| | RFCMH | Ours | 87.9 | 89.4 | 90.8 | 91.2 | 359.3 | **45.9** | **51.3** | **54.2** | **56.8** | **208.2** | **43.3** | **54.6** | **55.5** | **56.1** | **209.5** |

# B. Experimental Details

## B.1. Feature extraction

To ensure fair and reproducible comparisons, we follow the standard CMH benchmark protocol and use fixed, pre-extracted features for all datasets. For images, we adopt 4096-dimensional CNN activations from the second fully connected layer (fc7) of an ImageNet-pretrained backbone, which is typically based on AlexNet- or VGG-style architectures in CMH evaluations (Krizhevsky et al., 2012; Simonyan & Zisserman, 2014). For texts, we use two commonly adopted representations depending on the dataset release: sparse bag-of-words vectors with TF–IDF-style weighting, and dense 300-dimensional document embeddings produced by the Paragraph Vector (Doc2Vec) model (Le & Mikolov, 2014). All features are kept frozen during training, and all hashing methods are learned on top of the same feature inputs.

## B.2. Baselines Details

For a comprehensive comparison, we adopted the following baselines:

**General baselines.**

- **WASH** (Weakly-supervised enhAnced Semantic-aware Hashing) (Zhang et al., 2022a) simultaneously estimates label noise under weak supervision and performs semantic-aware hashing to improve cross-modal retrieval.

- **HCCH** (Hierarchical Correlation Cross-Modal Hashing) (Sun et al., 2023) exploits coarse-to-fine hierarchical semantics and learns hash codes whose Hamming distances preserve hierarchical relations for cross-modal retrieval.

- **HMAH** (Hierarchical Message Aggregation Hashing) (Tan et al., 2022) adopts an efficient teacher-student framework with hierarchical message aggregation to enhance cross-modal alignment and hashing learning.

- **MIAN** (Modality-Invariant Asymmetric Networks) (Zhang et al., 2022b) jointly preserves asymmetric intra-modal and inter-modal similarities and enforces modality-invariant representations for effective cross-modal hashing.

*Table 7.* The mAP scores under asymmetric label noise with a noise rate of 0.4 on three benchmark datasets. The best results are highlighted in **boldface** and the second-best are underlined. The column **Sum** indicates the total score of all hash lengths.

| Task | Method | Datasets Ref. | Xmedia | | | | | INRIA-Websearch | | | | | XmediaNet | | | | |
|---|---|---|---|---|---|---|---|---|---|---|---|---|---|---|---|---|---|
| | | | 16 | 32 | 64 | 128 | Sum | 16 | 32 | 64 | 128 | Sum | 16 | 32 | 64 | 128 | Sum |
| I2T | NRCH | MM'24 | 85.8 | 87.4 | 86.5 | 82.8 | 342.5 | 32.9 | 38.8 | 41.9 | 43.7 | 157.3 | 9.5 | 14.1 | 17.7 | 20.8 | 62.1 |
| | RSHNL | AAAI'25 | 79.6 | 79.5 | 79.0 | 79.7 | 317.8 | 34.4 | 43.6 | 48.3 | 48.1 | 174.4 | 34.3 | 44.6 | 50.0 | 51.5 | 180.4 |
| | NACD | NeurIPS'25 | 81.5 | 87.4 | 89.6 | 89.3 | 347.8 | 30.3 | 35.4 | 37.7 | 40.5 | 143.9 | 1.5 | 7.1 | 8.4 | 12.3 | 29.3 |
| | SCBCH | AAAI'26 | 33.9 | 79.5 | 85.8 | 86.6 | 285.8 | 6.4 | 18.0 | 34.2 | 46.6 | 105.2 | 1.3 | 8.7 | 15.7 | 28.5 | 54.2 |
| | RFCMH | Ours | **88.9** | **91.0** | **90.5** | **91.0** | **361.4** | **47.7** | **52.1** | **52.7** | **53.5** | **206.0** | **39.7** | **50.2** | **53.0** | **54.0** | **196.9** |
| T2I | NRCH | MM'24 | 85.3 | 86.9 | 87.3 | 83.8 | 343.3 | 32.3 | 39.7 | 43.4 | 45.4 | 160.8 | 10.9 | 15.8 | 20.1 | 24.2 | 71.0 |
| | RSHNL | AAAI'25 | 77.6 | 77.9 | 77.2 | 78.9 | 311.6 | 33.6 | 43.8 | 48.1 | 47.7 | 173.2 | 34.2 | 44.7 | 50.4 | 52.1 | 181.4 |
| | NACD | NeurIPS'25 | 81.6 | 87.1 | 90.5 | 90.3 | 349.5 | 29.2 | 35.5 | 38.5 | 41.5 | 144.7 | 1.6 | 8.5 | 10.3 | 14.4 | 34.8 |
| | SCBCH | AAAI'26 | 33.5 | 78.8 | 81.7 | 86.1 | 280.1 | 7.0 | 16.5 | 29.2 | 40.6 | 93.3 | 1.3 | 9.7 | 16.5 | 29.2 | 56.7 |
| | RFCMH | Ours | **88.0** | **89.7** | 88.9 | **90.3** | **356.9** | **45.6** | **51.0** | **53.0** | **53.8** | **203.4** | **40.6** | **49.8** | **53.3** | **54.4** | **198.1** |

*Table 8.* Additional ablation study under the clean setting.

| Task | Method | XMedia | | | | INRIA-Websearch | | | | XMediaNet | | | |
|---|---|---|---|---|---|---|---|---|---|---|---|---|---|
| | | 16 | 32 | 64 | 128 | 16 | 32 | 64 | 128 | 16 | 32 | 64 | 128 |
| I2T | w/o FAR | 89.2 | 91.1 | 91.3 | **91.8** | 43.2 | 48.8 | 50.1 | 49.5 | 40.1 | 48.1 | 55.1 | **56.0** |
| | w/o DGSA | 79.4 | 84.8 | 88.9 | 91.0 | 23.3 | 42.3 | 45.0 | 50.0 | 0.9 | 30.9 | 42.1 | 53.0 |
| | RFCMH (Ours) | **90.2** | **91.3** | **92.1** | 91.4 | **48.4** | **52.0** | **54.0** | **55.8** | **43.7** | **55.5** | **55.9** | **56.0** |
| T2I | w/o FAR | 87.1 | 88.8 | 89.0 | 90.4 | 42.1 | 48.5 | 50.6 | 49.5 | 41.0 | 48.5 | 55.0 | 55.8 |
| | w/o DGSA | 77.0 | 85.3 | 88.6 | 91.2 | 21.7 | 42.0 | 46.3 | 52.0 | 0.9 | 31.9 | 42.5 | 51.3 |
| | RFCMH (Ours) | **87.9** | **89.4** | **90.8** | **91.2** | **45.9** | **51.3** | **54.2** | **56.8** | **43.3** | **54.6** | **55.5** | **56.1** |

**Noise-robust baselines.**

- **CMMQ** (Cross-Modal Mutual Quantization) (Yang et al., 2022) combats noisy supervision by mutual quantization across modalities and proxy-based contrastive learning to stabilize hashing optimization.

- **DHRL** (Deep Hashing with Ranking Learning) (Shu et al., 2024) improves robustness under noisy labels via ranking-based learning with uncertainty-aware refinement for semantic alignment.

- **DHaPH** (Deep Hierarchy-aware Proxy Hashing) (Huo et al., 2024) constructs hierarchy-aware proxies and employs self-paced learning from easy to hard, providing indirect robustness to outliers and noisy supervision through stabilized proxy alignment.

- **RSHNL** (Robust Self-paced Hashing with Noisy Labels) (Pu et al., 2025c) incorporates self-paced learning to progressively train cross-modal hashing models from easy to hard samples under noisy labels.

- **DECH** (Deep Evidential Cross-modal Hashing) (Li et al., 2025b) introduces evidential modeling to quantify retrieval reliability while learning discriminative cross-modal hash codes.

- **NRCH** (Noise-Robust Cross-modal Hashing) (Wang et al., 2024) leverages a robust contrastive hashing loss with dynamic noise separation to reduce noise overfitting during training.

- **NACD** (Neighbor-aware Contrastive Disambiguation) (Su et al., 2026b) addresses redundant or ambiguous annotations by exploiting neighborhood agreement to disambiguate unreliable positives in contrastive hashing.

- **SCBCH** (Semantic-Consistent Bidirectional Contrastive Hashing) (Peng et al., 2025) leverages cross-modal semantic consistency and bidirectional soft contrastive learning to construct reliable supervision under noisy multi-label annotations.

# C. Extended Empirical Results and Analysis

### C.1. Extended Retrieval Results

To further evaluate the retrieval performance of RFCMH under the clean setting, we conduct experiments on three benchmark datasets, i.e., XMedia, INRIA-Websearch, and XMediaNet. Table 6 reports the mAP results under the clean setting on both

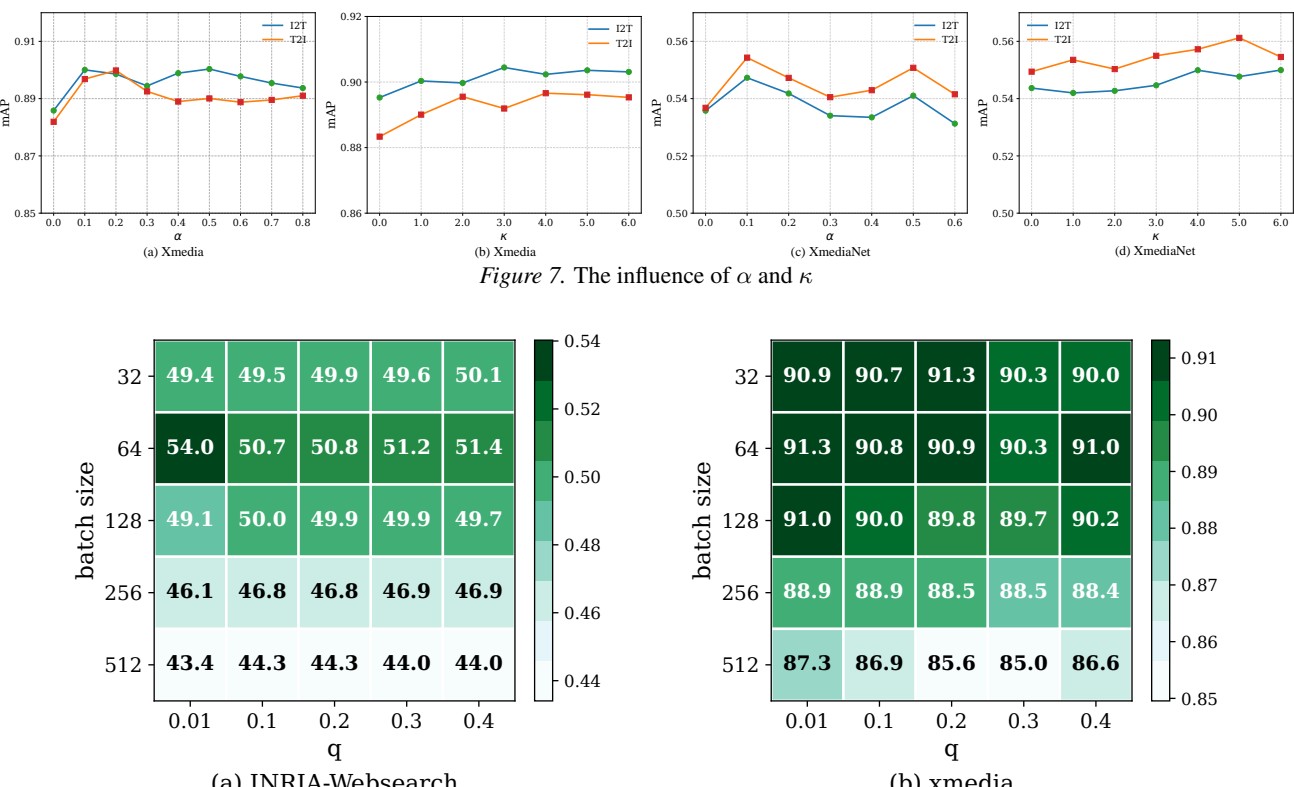

Figure 7. The influence of $\alpha$ and $\kappa$

Figure 8. Visualization of parameter sensitivity to batch size and $q$ on retrieval mAP.

I2T and T2I tasks. The results show that RFCMH achieves stable and competitive performance across different datasets and hash code lengths.

To further verify the applicability of our method under asymmetric noise, we additionally compare RFCMH with recent representative baselines and report the results under the 0.4 asymmetric noise rate, as shown in Table 7. The results indicate that RFCMH consistently outperforms most existing methods across both I2T and T2I tasks, highlighting its robustness and broad applicability across different noise levels and datasets.

To comprehensively evaluate the retrieval characteristics of the model under severe label noise, we plot precision–recall curves with 32-bit and 64-bit hash codes on the INRIA-Websearch, XMedia, and XMediaNet datasets under the 0.8 noise rate. As illustrated in Fig. 9, compared with existing methods, RFCMH maintains higher precision over a wide range of recall levels, demonstrating stronger discriminative capability and more stable retrieval performance under noisy supervision.

### C.2. Additional Ablation Analysis

To more clearly identify the source of the performance gains of RFCMH under the noise-free setting, we supplemented the ablation experiments to analyze the structural enhancement brought by DGSA and the actual contribution of FAR in the noise-free scenario. As shown in Table 8, under the noise-free setting, RFCMH achieves the best results for most code lengths and both retrieval tasks across the three datasets, indicating that these improvements are primarily driven by the effectiveness of our overall design. Specifically, the performance drops more noticeably after removing DGSA, with the most significant degradation observed on INRIA-Websearch and XMediaNet, which shows that the structural enhancement introduced by DGSA remains the main source of performance improvement in the noise-free scenario. In contrast, although removing FAR also leads to an overall decline in performance, the magnitude is relatively smaller, suggesting that under the noise-free setting, FAR mainly serves to provide auxiliary regularization, rather than being the primary source of gain.

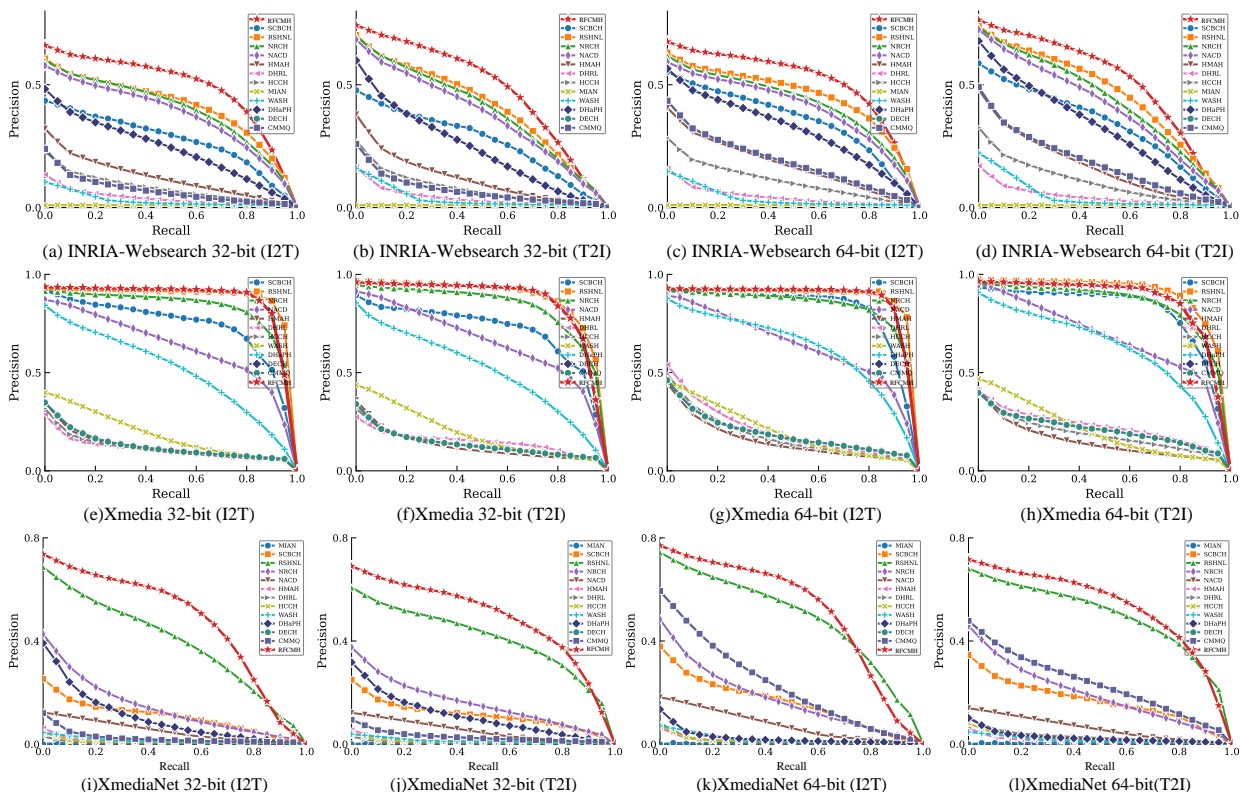

*Figure 9.* Precision-recall curves with different bit lengths on the INRIA-Websearch, Xmedia, and XmediaNet datasets, where (a-d), (e-h), and (i-l) show the curve results under 32 bits and 64 bits, respectively.

### C.3. Additional hyper-parameters.

As shown in Fig. 7, we use $\alpha = 0.5$ for XMedia and $\alpha = 0.1$ for XMediaNet and INRIA-Websearch. The retrieval performance consistently improves when $\kappa$ increases from 0 to 1 and peaks around $\kappa \in [3, 4]$, so we set $\kappa = 4$ in all experiments.

Fig. 8 further reports the sensitivity of retrieval mAP to the batch size and the robustness parameter $q$. We observe that moderate batch sizes consistently yield the best performance, whereas excessively large batches lead to noticeable degradation. For the robustness parameter $q$, the model exhibits more stable performance in the small-$q$ regime. Therefore, we fix $q = 0.01$ in all experiments to ensure stable optimization.

## D. Limitations and Future Work

Although RFCMH exhibits robustness against label noise, it currently lacks explicit mechanisms to handle cross-modal noisy correspondence. In practice, label noise and noisy correspondence often coexist, presenting complex challenges for reliable retrieval. Consequently, jointly modeling both label and correspondence uncertainties within a unified framework stands as a critical direction for future research.

