# OpenReview forum: "Learning with Admissibility: Robust Fuzzy Hashing for Cross-Modal Retrieval with Noisy Labels"
_ICML.cc/2026/Conference — ICML 2026 spotlight_

### Official Review · Reviewer_JxYC · 2026-03-05

**Soundness:** 4
**Presentation:** 3
**Significance:** 3
**Originality:** 3
**Overall Recommendation:** 5
**Confidence:** 4

**Summary:**

On the one hand, most existing CMH methods implicitly assume the availability of high-quality labeled multimodal data, ignoring the presence of noisy labels. On the other hand, current noise-resistant CMH methods typically discard predicted noisy instances or smooth discriminative signals to mitigate the impact of noisy labels. However, excessive separation leads to reduced data utilization, while smoothing weakens the ability to discriminate the true distribution of clean instances. To overcome these limitations, this paper proposes a novel robust fuzzy cross-modal hashing framework (RFCMH), which introduces fuzzy set theory to endow labels with acceptability, thereby obtaining reliable discriminative supervision from noisy labels.

**Compliance With Llm Reviewing Policy:**

Affirmed.

**Final Justification:**

The authors have solved all my concerns, and my assessment of this paper remains positive.

**Key Questions For Authors:**

A. Current experiments are primarily conducted under symmetric noise. Can the authors clarify whether RFCMH remains stable under asymmetric or instance-dependent noise? Additional empirical or theoretical discussion would strengthen the robustness claims.

B. FAR + DGSA introduce possibility/necessity estimation, reliability-based weighting, and dual-contrastive objectives. Compared with methods such as RSHNL or NRCH, how much additional training time and GPU memory does RFCMH require? A quantitative complexity analysis would improve practical transparency.

C. Under the noise-free setting, what primarily sustains the performance of RFCMH? Is the improvement mainly attributed to the structural enhancement from DGSA, or does the fuzzy modeling itself provide a regularization effect even without label noise? A clearer disentanglement of these contributions would enhance the understanding of the method.

**Limitations:**

yes

**Strengths And Weaknesses:**

**Strengths**

A. The paper presents a well-articulated motivation by identifying the limitations of existing noise-robust cross-modal hashing methods, particularly the drawbacks of hard noise separation and label smoothing strategies. The introduction of fuzzy set theory is logically positioned as a principled alternative.

B. The overall methodology is coherently designed. The integration of Fuzzy Admissibility Refinement (FAR) and Dual-Granularity Structural Alignment (DGSA) follows a structured optimization strategy, and the warm-up plus fine-tuning paradigm is technically reasonable.

C. The experiments are conducted across multiple benchmark datasets, different hash code lengths, and varying noise rates. The evaluation protocol is systematic, and comparisons with a broad set of baselines enhance the credibility of the empirical findings.

**Weaknesses**

A. The paper introduces fuzzy set theory into noisy CMH and constructs admissibility through possibility–necessity duality. The motivation is clear and conceptually reasonable. However, the definition of necessity in FAR (Eq. 1) and the admissibility aggregation scheme (Eq. 2) remain heuristic in nature. The paper lacks a deeper theoretical justification or formal analysis regarding the sufficiency or optimality of this formulation.

B. The experiments extensively evaluate performance under symmetric label noise (0.2–0.8), and the setup is comprehensive within this scope. However, more realistic noise patterns, such as asymmetric or instance-dependent noise, are not considered. This limits the practical generalizability of the conclusions.

C. The introduction of FAR and DGSA involves additional computations, including possibility–necessity estimation, reliability weighting, and bidirectional contrastive alignment. The manuscript does not explicitly analyze the additional training time or memory overhead compared to prior methods.

---

> ### Author Rebuttal · Authors · 2026-03-30
>
> Thanks for your valuable comments. Our responses are listed below.
>
> R-Question-1 To further examine the stability of RFCMH and strengthen the claim of robustness, we provide comparative experiments against representative methods; please refer to our response to Reviewer TbBE, R-Question-1.
>
> R-Question-2 To further enhance the practical transparency of RFCMH, we provide a systematic efficiency comparison against representative baseline methods, including per-epoch training time, GPU memory consumption, and model complexity. The detailed experimental settings, results, and discussion can be found in our response to Reviewer 6nGt, R-Question-3.
>
> R-Question-3 Thanks for your valuable comments. To more clearly identify the source of the performance gains of RFCMH under the noise-free setting, we supplemented the revised manuscript with corresponding ablation experiments to analyze the structural enhancement brought by DGSA and the actual contribution of FAR in the noise-free scenario. The detailed results are reported in the table below. Under the noise-free setting, RFCMH achieves the best results on most code lengths and both retrieval tasks across the three datasets, indicating that these improvements are likely also driven by the structural advantages and auxiliary regularization of our design. Specifically, the performance drops more noticeably after removing DGSA, with the most significant degradation observed on INRIA-Websearch and XMediaNet, which shows that the structural enhancement introduced by DGSA remains the main source of performance improvement in the noise-free scenario. In contrast, although removing FAR also leads to an overall decline in performance, the magnitude is relatively smaller, suggesting that under the noise-free setting, FAR mainly serves to stabilize training and provide auxiliary regularization, rather than being the primary source of gain.
>
> |Task|Method|Xmedia-16|Xmedia-32|Xmedia-64|Xmedia-128|INRIA-16|INRIA-32|INRIA-64|INRIA-128|XmediaNet-16|XmediaNet-32|XmediaNet-64|XmediaNet-128|
> |-|-|-|-|-|-|-|-|-|-|-|-|-|-|
> |I2T|w/o FAR|89.2|91.1|91.3|**91.8**|43.2|48.8|50.1|49.5|40.1|48.1|55.1|**56.0**|
> ||w/o DGSA|79.4|84.8|88.9|91.0|23.3|42.3|45.0|50.0|0.9|30.9|42.1|53.0|
> ||RFCMH|**90.2**|**91.3**|**92.1**|91.4|**48.4**|**52.0**|**54.0**|**55.8**|**43.7**|**55.5**|**55.9**|**56.0**|
> |T2I|w/o FAR|87.1|88.8|89.0|90.4|42.1|48.5|50.6|49.5|41.0|48.5|55.0|55.8|
> ||w/o DGSA|77.0|85.3|88.6|**91.2**|21.7|42.0|46.3|52.0|0.9|31.9|42.5|51.3|
> ||RFCMH|**87.9**|**89.4**|**90.8**|**91.2**|**45.9**|**51.3**|**54.2**|**56.8**|**43.3**|**54.6**|**55.5**|**56.1**|

---

> > ### Author Rebuttal · Reviewer_JxYC · 2026-04-03
> >
> > My issues have been addressed. I will keep my positive score.

---

> > > ### Author Response · Authors · 2026-04-04
> > >
> > > Thank you for your thoughtful review of our manuscript. We sincerely appreciate the time and effort you dedicated to evaluating our work and providing insightful comments.

---

### Official Review · Reviewer_6nGt · 2026-03-10

**Soundness:** 3
**Presentation:** 3
**Significance:** 3
**Originality:** 3
**Overall Recommendation:** 5
**Confidence:** 4

**Summary:**

The paper addresses the problem of label noise in Cross-Modal Hashing under real-world scenarios and proposes a novel Robust Fuzzy Cross-Modal Hashing framework (RFCMH). The proposed method first models noisy labels by leveraging possibility and necessity measures to characterize category-wise semantic admissibility. Subsequently, a Fuzzy Admissibility Refinement module is introduced to dynamically calibrate supervision signals, effectively preventing the model from being misled by false-positive samples. In addition, the framework incorporates a Dual-Granularity Structural Alignment strategy to enhance cross-modal alignment and instance-level uniformity, thereby ensuring stable and diverse learned representations. Experiments were conducted on three datasets, namely XMedia, INRIA-Websearch, and XMediaNet. The results demonstrated that RFCMH significantly outperformed 12 comparison methods under high-noise conditions, especially maintaining stable performance in high-noise scenarios.

**Compliance With Llm Reviewing Policy:**

Affirmed.

**Key Questions For Authors:**

1. Does the label-aware necessity measure in Equation (1) have a formal derivation or theoretical justification within fuzzy logic theory? Additionally, what is the specific motivation and functional role of introducing the parameter k in this formulation?
2. How does the admissibility distribution evolve during the training process? Specifically, does the gap between clean samples and noisy samples in terms of admissibility scores progressively widen as training proceeds?
3. The proposed RFCMH framework relies on a two-stage training strategy. However, the manuscript does not provide a theoretical time complexity analysis, nor does it report a direct comparison of training time with methods such as RSHNL. Could the authors provide additional information to clarify the computational efficiency of the method?
4. Under the clean setting, RFCMH performs relatively poorly on the XMedia dataset, while it still outperforms the baselines on the other two datasets. Could the authors provide an explanation for the reasons behind this discrepancy?

**Limitations:**

The method lacks rigorous complexity analysis, while the multi-stage training and extensive hyperparameter space potentially increase the computational cost and tuning difficulty.

**Strengths And Weaknesses:**

Strength
1. Introducing the possibility–necessity duality from fuzzy set theory into the field of cross-modal hashing represents an innovative perspective, as it allows for explicit modeling of semantic exclusivity within the cross-modal hashing framework.
2. Rather than directly imposing strong discriminative constraints under high-noise conditions, the framework adopts a two-stage strategy which aligns with the training dynamics of deep models by constructing supervision progressively.
3. Through mathematical proof, the authors demonstrate that the proposed q-contrastive loss exhibits a gradient growth that is asymptotically slower than traditional Log-loss as p→0, providing a solid theoretical justification for its superiority in suppressing outliers.
4. Extensive experimental design covering symmetric noise ratios from 0.2 to 0.8 reveals that RFCMH holds a distinct performance advantage over both general baselines and existing noise-robust methods, such as RSHNL and NRCH, particularly in extreme high-noise settings.

Weakness
1. The label-aware necessity measure in Equation (1) lacks a rigorous fuzzy logic derivation or theoretical proof. Additionally, the motivation and specific role of introducing the parameter k are not clearly explained.
2. Analysis regarding the evolution of the admissibility distribution during training is notably absent. It remains unclear whether the disparity in admissibility scores between clean and noisy samples gradually widens as the learning process progresses.
3. RFCMH employs a two-stage training process consisting of warm-up and fine-tuning. However, the manuscript does not include a formal analysis of the computational complexity or the overall training cost.
4. The proposed framework involves multiple key hyperparameters. Although sensitivity analyses are conducted for several parameters, the overall parameter space remains relatively large, which may increase the difficulty and cost of hyperparameter tuning in practice.

---

> ### Author Rebuttal · Authors · 2026-03-30
>
> Thanks for your constructive review. Our responses are listed below.
>
> R-Question-1 Thank you for your valuable comment. The necessity measure is theoretically grounded [1]. Specifically, for a fuzzy event $A$, this theory characterizes its uncertainty by possibility and necessity measures, where possibility ($\Pi(A)$) is used to measure the degree to which the evidence is compatible with $A$, and necessity ($N(A)$) is used to measure the degree to which the evidence necessarily supports $A$. These two measures satisfy the following dual axiom: $$N(A) = 1 - \Pi(A^c)$$, indicating that the necessity of event $A$ is equal to the degree to which its complement $A^c$ is impossible. In our framework, when the class label is annotated as k, the necessity measure for class k is equivalent to how small the possibility is that it does not belong to class c. Therefore, it can be represented by the maximum possibility that the sample belongs to any class other than $k$. When the class is not annotated as k, the necessity of the sample belonging to class $k$ logically degenerates into the observed possibility-based verification.
>
> For parameter κ, under clean labels, necessity can be characterized by the maximum possibility among non-target classes. Under noisy labels, however, directly using this form may incorporate noise-perturbed yet semantically plausible classes into optimization and thus introduce bias. We therefore use a Top-κ aggregation strategy so that necessity is jointly characterized by multiple high-response non-target classes rather than a single class response, which reduces sensitivity to noisy labels and improves stability. Parameter sensitivity analyses in the main text and appendix show the best performance at κ=3 or 4, with an average gain of about 1 percentage point.
>
> [1] Liu B. Uncertainty theory[M]//Uncertainty theory: A branch of mathematics for modeling human uncertainty. Berlin, Heidelberg: Springer Berlin Heidelberg, 2007: 1-79.
>
> R-Question-2 To show the dynamic changes of acceptability scores during training, we provide a visualization of the weight distribution under noisy labels at https://anonymous.4open.science/r/RFCMH-F105/distribution.png. As training proceeds, clean samples are progressively assigned higher weights, while noisy samples remain in the low-weight region. This shows that the proposed mechanism gradually emphasizes reliable supervision and suppresses noisy interference, leading to more stable and discriminative representation learning.
>
> R-Question-3 Thank you for the suggestion. To clarify the computational efficiency of RFCMH, we provide a systematic comparison with representative baselines in terms of per-epoch training time, GPU memory consumption, and model complexity. All methods were evaluated under the same batch size. Despite its two-stage strategy, RFCMH remains competitive in overall computational cost.
>
> | Method | Epoch↓ | Alloc.↓ | Reserv.↓ | Params(M)↓ | Size↓ |
> |---|---:|---:|---:|---:|---:|
> | NACD | 30.88 | 3590.33 | 4928 | 111.00 | 423.44 |
> | RSHNL | **2.04** | 1566.61 | 1582 | 73.21 | 279.28 |
> | SCBCH | 32.38 | 3998.36 | 4062 | 111.00 | 423.44 |
> | RFCMH | 2.72 | **1294.69** | **1318** | **55.49** | **211.69** |
>
> R-Question-4 Thank you for your suggestion. We also noticed this phenomenon during our experiments. We believe that this result is mainly related to the category structure of the XMedia dataset itself, as well as the way DGSA $L_{\mathrm{inter}}$ works. Specifically, $L_{\mathrm{inter}}$ is originally designed for noisy-label scenarios, where it suppresses erroneous aggregation under noisy supervision by imposing an additional separation constraint on samples without label intersection. To explain its rationality, we consider the single-label noisy case here. Let the number of classes be $C$, and the noise ratio be $\eta$. For a pair of samples with different true classes, the probability that their observed labels remain the same after noise perturbation is
> $$P(\tilde{y}_i = \tilde{y}_j \mid y_i \neq y_j)=\frac{2\eta}{C-1}-\frac{C\eta^2}{(C-1)^2}\leq\frac{2\eta}{C-1}.$$
> Accordingly, we have
> $$P(\tilde{y}_i \neq \tilde{y}_j \mid y_i \neq y_j)=1-\frac{2\eta}{C-1}+\frac{C\eta^2}{(C-1)^2}\geq1-\frac{2\eta}{C-1}.$$
> This indicates that when the number of classes $C$ is relatively large,
>
> $$ P(\tilde{y}_i \neq \tilde{y}_j \mid y_i \neq y_j) \gg P(\tilde{y}_i = \tilde{y}_j \mid y_i \neq y_j). $$
>
> Therefore, under symmetric noise, imposing a separation constraint on samples without label intersection is generally reasonable and effective. However, XMedia contains only 20 classes, and its category partition is relatively coarse-grained, so there often still exists strong latent semantic relevance among different labels. In this case, the separation constraint imposed by $L_{\mathrm{inter}}$ on samples without label intersection may be slightly too strict, and therefore may not always be optimal under the clean-label setting.

---

### Official Review · Reviewer_TbBE · 2026-03-12

**Soundness:** 3
**Presentation:** 3
**Significance:** 3
**Originality:** 3
**Overall Recommendation:** 5
**Confidence:** 4

**Summary:**

To address the negative impact of noisy labels, existing CMH methods follow two paradigms: noise separation and label smoothing. However, excessive separation leads to reduced data utilization, while smoothing weakens the ability to discriminate the true distribution of clean instances. Therefore, this paper proposes a novel robust fuzzy cross-modal hashing framework (RFCMH) that incorporates fuzzy set theory to impart label acceptability, thereby obtaining reliable discriminative supervision from noisy labels.

**Compliance With Llm Reviewing Policy:**

Affirmed.

**Key Questions For Authors:**

Q1：What is the fundamental difference between symmetric noise and asymmetric noise? Since the experiments are primarily conducted under symmetric noise settings, the authors are encouraged to clarify the distinction between these two types of noise and discuss the applicability of the proposed method under asymmetric noise scenarios.

Q2：It would be beneficial to further validate the effectiveness of fuzzy set theory in handling label noise. For instance, more systematic comparisons with representative noise separation strategies and label smoothing methods could help clarify where the performance gains originate.

Q3：Several baselines (e.g., MIAN, DHRL, and SCBCH) exhibit severe performance degradation under high noise rates. Could the authors provide a more detailed analysis of why these methods fail in noisy settings? For example, are they inherently sensitive to noisy supervision, or do they rely heavily on the assumption of clean labels? A deeper discussion would enhance the interpretability and analytical strength of the paper.

Q4: The paper introduces the idea of incorporating more matching pairs in the DGSA module to improve structural alignment. However, the underlying rationale for this decision is not sufficiently explained, and it remains unclear whether the additional matching pairs can effectively enhance the model's performance. The authors are encouraged to provide further clarification on the principles behind this design choice.

**Limitations:**

yes

**Strengths And Weaknesses:**

Strengths:
1. Good originality. This paper introduces a Robust Fuzzy Cross-Modal Hashing (RFCMH) framework from the perspective of fuzzy set theory. By incorporating the concept of admissibility into supervision modeling, the method effectively mitigates the adverse impact of noisy labels on discriminative learning.
2. Comprehensive experimental validation. The authors conduct extensive experiments on multiple benchmark datasets under various noise rates. Comparisons with a wide range of state-of-the-art methods demonstrate the effectiveness and superiority of the proposed approach. The empirical evaluation is thorough and generally convincing.

Weaknesses:
1. The formatting of references is not fully consistent. In particular, the capitalization of some conference and journal names varies throughout the bibliography. The authors are encouraged to carefully revise the formatting to ensure consistency and professionalism.
2. Although fuzzy set theory serves as the core foundation of the proposed method, the paper lacks dedicated experiments that directly validate the contribution of the fuzzy modeling itself.
3. The paper introduces the idea of incorporating more matching pairs in the DGSA module to improve structural alignment. However, the underlying rationale for this decision is not sufficiently explained, and it remains unclear whether the additional matching pairs are reliably beneficial for enhancing the model's performance.

---

> ### Author Rebuttal · Authors · 2026-03-30
>
> We appreciate your valuable feedback. Our responses are listed below.
>
> R-Question-1 Symmetric noise flips the true label to any other class with equal probability and is usually unrelated to semantic structure. In contrast, asymmetric noise is more likely to occur between semantically similar or easily confused classes, and is therefore closer to real annotation scenarios. To further verify the applicability of our method under asymmetric noise, we additionally compare RFCMH with representative recent baselines and report the results under an asymmetric noise ratio of 0.4, as shown below.
>
> |Task|Method|Ref|Xmedia-16|Xmedia-32|Xmedia-64|Xmedia-128|INRIA-16|INRIA-32|INRIA-64|INRIA-128|XmediaNet-16|XmediaNet-32|XmediaNet-64|XmediaNet-128|
> |-|-|-|-|-|-|-|-|-|-|-|-|-|-|-|
> |I2T|NRCH|MM'24|85.8|87.4|86.5|82.8|32.9|38.8|41.9|43.7|9.5|14.1|17.7|20.8|
> ||RSHNL|AAAI'25|79.6|79.5|79.0|79.7|34.4|43.6|48.3|48.1|34.3|44.6|50.0|51.5|
> ||NACD|NeurIPS'25|81.5|87.4|89.6|89.3|30.3|35.4|37.7|40.5|1.5|7.1|8.4|12.3|
> ||SCBCH|AAAI'26|33.9|79.5|85.8|86.6|6.4|18.0|34.2|46.6|1.3|8.7|15.7|28.5|
> ||RFCMH|Ours|**88.9**|**91.0**|**90.5**|**91.0**|**47.7**|**52.1**|**52.7**|**53.5**|**39.7**|**50.2**|**53.0**|**54.0**|
> |T2I|NRCH|MM'24|85.3|86.9|87.3|83.8|32.3|39.7|43.4|45.4|10.9|15.8|20.1|24.2|
> ||RSHNL|AAAI'25|77.6|77.9|77.2|78.9|33.6|43.8|48.1|47.7|34.2|44.7|50.4|52.1|
> ||NACD|NeurIPS'25|81.6|87.1|**90.5**|**90.3**|29.2|35.5|38.5|41.5|1.6|8.5|10.3|14.4|
> ||SCBCH|AAAI'26|33.5|78.8|81.7|86.1|7.0|16.5|29.2|40.6|1.3|9.7|16.5|29.2|
> ||RFCMH|Ours|**88.0**|**89.7**|88.9|**90.3**|**45.6**|**51.0**|**53.0**|**53.8**|**40.6**|**49.8**|**53.3**|**54.4**|
>
> From the results, it can be seen that the proposed method does not rely on a specific noise form and can still maintain strong robust representation learning ability under asymmetric noise.
>
> R-Question-2 Thank you for your valuable suggestion. To further verify the effectiveness of fuzzy set theory in handling label noise, we additionally provide comparisons and analyses with representative noise separation methods and label smoothing methods. Please refer to our response to Reviewer hEFk, R-Question-1, for the relevant experimental results and discussion.
>
> R-Question-3 Thank you for your valuable comment. To explain why some baselines degrade severely under high noise ratios, we provide the following analysis.
>
> 1) MIAN relies on modality alignment and observed cross-modal similarity preservation, but lacks an explicit mechanism to suppress noisy supervision. Under high noise ratios, noisy annotations directly distort both alignment and similarity preservation, causing severe degradation.
>
> 2) DHRL separates clean and noisy samples according to loss magnitude. Under high noise ratios, however, the model may overfit noisy labels, causing some noisy samples to also obtain small losses. This weakens sample discrimination and leads to severe degradation.
>
> 3) SCBCH is designed for noisy multi-label learning with a soft contrastive mechanism. Under high-noise single-label settings, however, the constructed positive and negative pairs may contain many errors, introducing incorrect semantic alignment signals and severely degrading performance.
>
> R-Question-4 Thank you for your valuable suggestion. To further clarify the rationale behind introducing more matching relationships in DGSA, we provide the following explanation from two aspects.
>
> First, standard contrastive learning emphasizes an index-consistent paired alignment mechanism, where only diagonal sample pairs are treated as positive pairs, while all remaining samples are uniformly included in the denominator term as negatives. Although this design is beneficial for preserving strict one-to-one paired consistency, it often overlooks richer semantic relatedness in the non-diagonal region. In cross-modal retrieval scenarios, this inevitably causes some semantically related samples that are not explicitly labeled as positive pairs to be treated as negative samples, thereby introducing false negatives and limiting the model’s ability to capture broader global semantic structure.
>
> Second, under noisy-label supervision, to show that $L_{\mathrm{inter}}$ term is not a heuristic design but rather a structurally constrained term with a clear statistical basis, the detailed derivation can be found in our response to Reviewer 6nGt, R-Question-4.
>
> To more intuitively demonstrate the effect of this design, we provide visualization results at https://anonymous.4open.science/r/RFCMH-F105/visualization.png
> , comparing the similarity structure distributions of standard contrastive learning and the full DGSA. The results show that standard contrastive learning mainly concentrates high-similarity responses in the diagonal region, whereas DGSA significantly enhances the structural responses of semantically related samples in the non-diagonal region, indicating stronger global semantic alignment beyond strict one-to-one pairing.

---

### Official Review · Reviewer_hEFk · 2026-03-12

**Soundness:** 4
**Presentation:** 4
**Significance:** 3
**Originality:** 4
**Overall Recommendation:** 5
**Confidence:** 4

**Summary:**

This paper proposes a novel robust fuzzy cross-modal hashing framework (RFCMH) that incorporates fuzzy set theory to impart label acceptability, thereby obtaining reliable discriminative supervision from noisy labels. Specifically, RFCMH first models noisy labels using probability and necessity measures. Subsequently, Fuzzy Admissibility Refinement (FAR) is proposed to dynamically calibrate the supervision signal, thus preventing the model from being misled by false positives. Furthermore, Dual-Granularity Structural Alignment (DGSA) is introduced to enforce cross-modal alignment and instance-level consistency, thereby ensuring stable and diverse representations.

**Compliance With Llm Reviewing Policy:**

Affirmed.

**Final Justification:**

I choose to maintain my original score.

**Key Questions For Authors:**

1. It is recommended that the authors provide more explicit comparisons with representative noise separation and label smoothing approaches. Such experiments would more clearly highlight the advantages of the proposed fuzzy modeling framework.

2. The authors are encouraged to expand the related work section by incorporating relevant studies from the fuzzy computing and fuzzy learning literature, thereby strengthening the theoretical grounding of the paper.

3. The manuscript would benefit from a deeper discussion of why point estimation tends to produce overconfident predictions and how this overconfidence negatively affects representation learning in noisy-label settings. Theoretical analysis, illustrative examples, or empirical evidence would help clarify this point.

**Limitations:**

yes

**Strengths And Weaknesses:**

Strengths

1. The work addresses the realistic and important problem of noisy supervision in cross-modal hashing, which frequently occurs in large-scale multimedia retrieval scenarios. Therefore, the study has clear practical significance.

2. The proposed framework is well-structured, with clearly described methodological steps and training procedures. The overall presentation is easy to follow, making the approach relatively straightforward to reproduce. The paper demonstrates good readability.

3.  The proposed method achieves consistently superior performance compared to existing state-of-the-art (SOTA) approaches across multiple datasets and noise levels, demonstrating its effectiveness and robustness.

Weaknesses

1. Although the authors point out the limitations of existing noise-robust cross-modal hashing methods (e.g., noise separation and label smoothing), these claims are mainly presented in a qualitative manner. The lack of direct experimental evidence or controlled comparisons weakens the persuasiveness of the argument.

2. The related work section mainly focuses on cross-modal hashing methods, while overlooking relevant studies from the fuzzy computing community. Given that fuzzy set theory is central to the proposed framework, a broader discussion of related fuzzy learning approaches would strengthen the theoretical foundation.

3. The authors briefly state that deterministic point estimation leads to overconfidence and may harm representation learning under noisy labels. However, this claim is not thoroughly elaborated or theoretically justified. A more detailed explanation would improve conceptual clarity.

---

> ### Author Rebuttal · Authors · 2026-03-30
>
> Thanks for your valuable comments. Our responses are listed below.
>
> R-Question-1: To highlight the advantage of the proposed fuzzy framework, we further compare it with representative noise separation and label smoothing methods.
>
> Specifically, for noise separation methods, we have provided a t-SNE comparison between our method and NRCH in Appendix C.3, Figure 9 of the uploaded manuscript. As shown there, NRCH exhibits obvious clustering collapse, with samples concentrated in a dense and nearly indistinguishable region, especially those identified as suspected noisy samples. In contrast, RFCMH still preserves relatively clear structural separation. This suggests that relying solely on hard noise separation is insufficient to characterize the semantic features of such samples, further demonstrating the advantage of the proposed fuzzy framework.
>
> For label smoothing methods, we visualize the sample weight distributions on INRIA-Websearch under a noise ratio of 0.2 at both the early stage of training and Epoch 30, as shown at https://anonymous.4open.science/r/RFCMH-F105/Comparison.png. SCBCH assigns lower weights to many originally clean samples, causing substantial overlap between clean and noisy samples and weakening their separability. In contrast, RFCMH maintains a clearer distribution throughout training, preserving the boundary between clean and noisy samples while avoiding unnecessary suppression of reliable supervision. This suggests that RFCMH suppresses unreliable samples more selectively, leading to more robust representation learning.
>
> R-Question-2 We thank the reviewer for the suggestion. We summarize representative studies from fuzzy computing and fuzzy learning as follows. Zadeh’s fuzzy set theory provides the basis for modeling semantic relations through graded membership rather than rigid binary assignment [1]. Classical fuzzy learning methods such as fuzzy c-means further show that soft membership is well suited for handling overlap, ambiguity, and noisy observations [2]. In addition, neuro-fuzzy systems and recent deep models with fuzzy mechanisms demonstrate that fuzzy reasoning can be incorporated into trainable representation learning frameworks [3,4]. Different from these studies, our method is designed for noisy cross-modal hashing and develops a fuzzy semantic modeling framework for robust cross-modal retrieval under label noise.
>
> [1] Zadeh L A. Fuzzy sets[J]. Information and control, 1965, 8(3): 338-353.
>
> [2] Bezdek J C, Ehrlich R, Full W. FCM: The fuzzy c-means clustering algorithm[J]. Computers & geosciences, 1984, 10(2-3): 191-203.
>
> [3] Jang J S R. ANFIS: adaptive-network-based fuzzy inference system[J]. IEEE transactions on systems, man, and cybernetics, 1993, 23(3): 665-685.
>
> [4] Nan Y, Del Ser J, Tang Z, et al. Fuzzy attention neural network to tackle discontinuity in airway segmentation[J]. IEEE transactions on neural networks and learning systems, 2023, 35(6): 7391-7404.
>
> R-Question-3 Thank you for your valuable comment. To more clearly illustrate the limitations of point-estimate supervision under noisy labels, as well as the role of the necessity constraint in suppressing the continuous reinforcement of erroneous relations, we further analyze that point-estimate supervision methods typically compress the relationship between a sample and its categories into a single deterministic target, and continuously optimize toward this target during training. In noisy-label scenarios, once this target is corrupted by mislabeled annotations, the incorrect semantic relations may be repeatedly reinforced in the form of fixed supervisory signals, thereby causing the distributions of clean and noisy samples in the weight space to gradually overlap, and further undermining the discriminability and structural stability of cross-modal representations.
>
> As shown at https://anonymous.4open.science/r/RFCMH-F105/Point.png, we compare the sample weight distributions on INRIA-Websearch between the variant using only possibility as the label weight and the full RFCMH under a noise ratio of 0.2 at Epoch 30. When only possibility is used, many clean samples are distributed across the full interval from 0 to 1 and substantially overlap with noisy samples. This suggests that possibility alone is insufficient to distinguish truly reliable samples from those affected by noisy labels, and may continuously preserve erroneous relations. In contrast, the full RFCMH exhibits a clearer distribution with a more distinct separation between clean and noisy samples. This is because RFCMH does not directly optimize a high possibility value as a fixed supervisory signal, but further incorporates necessity to constrain category exclusiveness, preventing samples with high acceptability but clear inter-class conflicts from being repeatedly reinforced. As a result, RFCMH suppresses the interference of noisy labels more effectively, leading to more robust cross-modal representations.

---

> > ### Author Rebuttal · Reviewer_hEFk · 2026-04-02
> >
> > My questions have been well resolved. I would like to maintain my current score unchanged.

---

> > > ### Author Response · Authors · 2026-04-03
> > >
> > > We sincerely appreciate your positive feedback. We are pleased that our response has addressed your concerns, and we are grateful for your recognition and support of our work.

---

### Decision · Program_Chairs · 2026-04-30

**Decision:**

Accept (spotlight)

**Comment:**

Thank you for your submission.

The reviewers reached a unanimous consensus (5, 5, 5, 5), highlighting the paper as a technically solid contribution with strong empirical/theoretical support. The rebuttal period confirmed the robustness of the proposed method, and no significant concerns remain.